# SCHEDULE-ROBUST ONLINE CONTINUAL LEARNING

## ABSTRACT

A continual learning (CL) algorithm learns from a non-stationary data stream. The non-stationarity is modeled by some schedule that determines how data is presented over time. Most current methods make strong assumptions on the schedule and have unpredictable performance when such requirements are not met. A key challenge in CL is thus to design methods robust against arbitrary schedules over the same underlying data, since in real-world scenarios schedules are often unknown and dynamic. In this work, we introduce the notion of *schedule-robustness* for CL and a novel approach satisfying this desirable property in the challenging online class-incremental setting. We also present a new perspective on CL, as the process of learning a schedule-robust predictor, followed by adapting the predictor using only replay data. Empirically, we demonstrate that our approach outperforms existing methods on CL benchmarks for image classification by a large margin.

## 1 INTRODUCTION

A hallmark of natural intelligence is its ability to continually absorb new knowledge while retaining and updating existing one. Achieving this objective in machines is the goal of continual learning (CL). Ideally, CL algorithms learn online from a never-ending and non-stationary stream of data, without catastrophic forgetting (McCloskey & Cohen, 1989; Ratcliff, 1990; French, 1999).

The non-stationarity of the data stream is modeled by some *schedule* that defines what data arrives and how its distribution evolves over time. Two family of schedules commonly investigated are *task-based* (De Lange et al., 2021) and *task-free* (Aljundi et al., 2019b). The task-based setting assumes that new data arrives one task at a time and data distribution is stationary for each task. Many CL algorithms (e.g., Buzzega et al., 2020; Kirkpatrick et al., 2017; Hou et al., 2019) thus train *offline*, with multiple passes and shuffles over task data. The task-free setting does not assume the existence of separate tasks but instead expects CL algorithms to learn *online* from streaming data, with evolving sample distribution (Caccia et al., 2022; Shanahan et al., 2021). In this work, we tackle the task-free setting with focus on class-incremental learning, where novel classes are observed incrementally and a single predictor is trained to discriminate all of them (Rebuffi et al., 2017).

Existing works are typically designed for specific schedules, since explicitly modeling and evaluating across all possible data schedules is intractable. Consequently, methods have often unpredictable performance when scheduling assumptions fail to hold (Farquhar & Gal, 2018; Mundt et al., 2022; Yoon et al., 2020). This is a considerable issue for practical applications, where the actual schedule is either unknown or may differ from what these methods were designed for. This challenge calls for an ideal notion of *schedule-robustness*: CL methods should behave consistently when trained on different schedules over the same underlying data.

To achieve schedule-robustness, we introduce a new strategy based on a two-stage approach: 1) learning online a schedule-robust predictor, followed by 2) adapting the predictor using only data from experience replay (ER) (Chaudhry et al., 2019b). We will show that both stages are robust to diverse data schedules, making the whole algorithm schedule-robust. We refer to it as **SC**hedule-**R**obust **O**nline continua**L** **L**earning (SCROLL). Specifically, we propose two online predictors that by design are robust against arbitrary data schedules and catastrophic forgetting. To learn appropriate priors for these predictors, we present a meta-learning perspective (Finn et al., 2017; Wang et al., 2021) and connect it to the pre-training strategies in CL (Mehta et al., 2021). We show that pre-training offers an alternative and efficient procedure for learning predictor priors instead of directly solving the meta-learning formulation. This makes our method computationally competitive and at the same

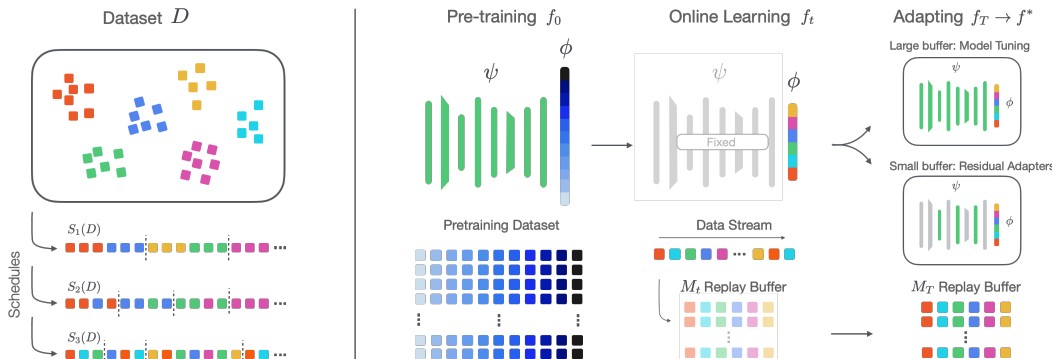

**Figure 1: Left.** Illustration of a classification dataset $D$ streamed according to different schedules (dashed vertical lines identify separate batches). **Right.** Pre-training + the two stages of SCROLL: 1) online learning and store replay samples from data stream, 2) adapting the predictor using the replay buffer (green indicates whether the representation $\psi$ is being updated).

time offers a clear justification for adopting pre-training in CL. Finally, we present effective routines for adapting the predictors from the first stage. We show that using only ER data for this step is key to preserving schedule-robustness, and discuss how to mitigate overfitting when ER data is limited.

**Contributions.** 1) We introduce the novel concept of schedule-robustness for CL, an important property that lacks in existing methods. 2) We propose a novel online strategy that satisfies schedule-robustness, along with practical algorithms. 3) Theoretically, we connect CL to standard meta-learning methods via schedule-robustness. This justifies the use of pre-trained models as a knowledge-prior for CL. We further show that multi-class classification is an efficient and principled procedure for learning such priors in CL. 4) Empirically, we show that SCROLL outperforms state-of-the-art methods by a large margin (over a 20% improvement in accuracy in many settings). This further supports the focus of this work on schedule-robustness and our strategy to achieve it in practice.

## 2 PRELIMINARIES AND RELATED WORKS

We formalize CL as learning from non-stationary data sequences. A data sequence consists of a dataset $D = \{(x_i, y_i)\}_{i=1}^N$ regulated by a **schedule** $S = (\sigma, \beta)$. Applying the schedule $S$ to $D$ is denoted by $S(D) \triangleq \beta(\sigma(D))$, where $\sigma(D)$ is a specific ordering of $D$, and $\beta(\sigma(D)) = \{B_t\}_{t=1}^T$ splits the sequence $\sigma(D)$ into $T$ batches of samples $B_t = \{(x_{\sigma(i)}, y_{\sigma(i)})\}_{i=k_t}^{k_{t+1}}$, with $k_t$ the batch boundaries. Intuitively, $\sigma$ determines the order in which $(x, y) \in D$ are observed, while $\beta$ determines how many samples are observed at a time. Fig. 1 (Left) illustrates how the same dataset $D$ could be streamed according to different schedules. For example, $S_1(D)$ in Fig. 1 (Left) depicts the standard schedule to split and stream $D$ in batches of $C$ classes at the time ($C = 2$).

### 2.1 CONTINUAL LEARNING

A CL algorithm learns from $S(D)$ one batch $B_t$ at a time, iteratively training a predictor $f_t : \mathcal{X} \to \mathcal{Y}$ to fit the observed data. Some formulations assume access to a fixed-size *replay buffer* $M$, which mitigates forgetting by storing and reusing samples for future training. Given an initial predictor $f_0$ and an initial buffer $M_0$, we define the update rule of a CL algorithm $\text{Alg}(\cdot)$ at step $t$ as

$$(f_t, \ M_t) = \text{Alg}(B_t, f_{t-1}, M_{t-1}), \tag{1}$$

where the algorithm learns from the current batch $B_t$ and updates both the replay buffer $M_{t-1}$ and predictor $f_{t-1}$ from the previous iteration.

At test time, the performance of the algorithm is evaluated on a distribution $\pi_D$ that samples $(x, y)$ sharing the same labels with the samples in $D$. The generalization error is denoted by

$$\mathcal{L}(S(D), f_0, \text{Alg}) = \mathbb{E}_{(x,y) \sim \pi_D} \ell(f_T(x), y) \tag{2}$$

where the final predictor $f_T$ is recursively obtained from (1) and $f_0$.[1] In the following, we review existing approaches for updating $f_t$ and $M_t$, and strategies for initializing $f_0$.

**Predictor Update ($f_t$).** Most CL methods learn $f_t$ by solving an optimization problem of the form:

$$f_t = \arg\min_{f} \quad \alpha_1 \cdot \underbrace{\sum_{(x,y) \in B_t} \ell(f(x), y)}_{\text{current batch loss}} + \alpha_2 \cdot \underbrace{\sum_{(x,y) \in M_{t-1}} \ell(f(x), y)}_{\text{replay loss}} + \alpha_3 \cdot \underbrace{R(f, f_{t-1})}_{\text{regularization loss}} \quad (3)$$

where $\alpha_{1,2,3}$ are prescribed non-negative weights, $\ell$ a loss function, and $R$ a regularizer. This general formulation for updating $f_t$ recovers replay-based methods such as iCarl (Rebuffi et al., 2017) and DER (Buzzega et al., 2020) for specific choices of $\ell$ and $R$. Moreover, if $M_t = \varnothing \; \forall t$ (or we set $\alpha_2 = 0$), the replay loss is omitted and we recover regularization-based methods (e.g Kirkpatrick et al., 2017; Li & Hoiem, 2017; Yu et al., 2020) that update selective parameters of $f_{t-1}$ to prevent forgetting.

**Replay Buffer Update ($M_t$).** In replay-based methods, Alg($\cdot$) must also define a *buffering strategy* that decides what samples to store in $M_t$ for future reuse, and which ones to discard from a full buffer. Common strategies include exemplar selection (Rebuffi et al., 2017) and random reservoir sampling (Vitter, 1985), with more sophisticated methods like gradient-based sample selection (Aljundi et al., 2019c) and meta-ER (Riemer et al., 2019). As in (3), replay-based methods typically mix the current data with replay data for predictor update (e.g Aljundi et al., 2019c; Riemer et al., 2019; Caccia et al., 2022). In contrast, Prabhu et al. (2020) learns the predictor using only replay data. We will adopt a similar strategy and discuss how it is crucial for achieving schedule-robustness.

**Predictor Initialization ($f_0$).** The initial predictor $f_0$ in (2) represents the prior knowledge available to CL algorithms, before learning from sequence $S(D)$. Most methods are designed for randomly initialized $f_0$ with no prior knowledge (e.g., Rebuffi et al., 2017; Gupta et al., 2020; Prabhu et al., 2020; Kirkpatrick et al., 2017). However, this assumption may be overly restrictive for several applications (e.g., vision-based tasks like image classification), where available domain knowledge and data can endow CL algorithms with more informative priors than random initialization. We review two strategies for predictor initialization relevant to this work.

*Initialization by Pre-training.* One way for initializing $f_0$ is to pre-train a representation on data related to the CL task (e.g., ImageNet for vision-based tasks) via either self-supervised learning (Shanahan et al., 2021) or multi-class classification (Mehta et al., 2021; Wang et al., 2022; Wu et al., 2022). Boschini et al. (2022) observed that while pre-training mitigates forgetting, model updates quickly drift the current $f_t$ away from $f_0$, diminishing the benefits of prior knowledge as CL algorithms continuously learn from more data. To mitigate this, Shanahan et al. (2021); Wang et al. (2022) keep the pre-trained representation fixed while introducing additional parameters for learning the sequence. In contrast, we will offer effective routines for updating the pre-trained representation and significantly improve test performance.

*Initialization by Meta-Learning.* Another approach for initializing $f_0$ is meta-learning (Hospedales et al., 2021). Given the CL generalization error in (2), we may learn $f_0$ by solving the meta-CL problem below,

$$f_0 = \arg\min_{f} \quad \mathbb{E}_{(D,S) \sim \mathcal{T}} \; \mathcal{L}(S(D), f, \text{Alg}) \quad (4)$$

where $\mathcal{T}$ is a meta-distribution over datasets $D$ and schedules $S$. For instance, Javed & White (2019) set Alg($\cdot$) to be MAML (Finn et al., 2017) and observed that the learned $f_0$ encodes sparse representation to mitigate forgetting. However, directly optimizing (4) is computationally expensive since the cost of gradient computation scales with the size of $D$. To overcome this, we will leverage Wang et al. (2021) to show that meta-learning $f_0$ is analogous to pre-training for certain predictors, which provides a much more efficient procedure to learn $f_0$ without directly solving (4).

## 2.2 SCHEDULE-INVARIANCE AND SCHEDULE-ROBUSTNESS

The majority of CL methods depend significantly on the specific data schedule. This typically leads to unpredictable behaviors when considering different schedules (See e.g. Farquhar & Gal, 2018; Yoon et al., 2020; Mundt et al., 2022, or our experiments in Sec. 4.1). To address this issue,

---

[1] We omitted the initial memory buffer $M_0$ since it is typically empty.

we first introduce the ideal notion of schedule-invariance for CL. We say that a CL algorithm is **schedule-invariant** if

$$\mathcal{L}(S_1(D), f_0, \text{Alg}) = \mathcal{L}(S_2(D), f_0, \text{Alg}), \qquad \forall S_1, S_2 \text{ schedules}, \ \forall D \text{ dataset}. \tag{5}$$

Eq. (5) captures the idea that CL algorithms should perform *identically* for all data schedules. In practice, however, inherent randomness of learning algorithms often leads to similar but not identical performance. We thus further introduce **schedule-robustness** as a relaxation of schedule-invariance:

$$\mathcal{L}(S_1(D), f_0, \text{Alg}) \approx \mathcal{L}(S_2(D), f_0, \text{Alg}), \qquad \forall S_1, S_2 \text{ schedules}, \ \forall D \text{ dataset}, \tag{6}$$

where the $\approx$ symbol loosely captures potentially small differences in performance[2]. We argue that achieving schedule-robustness is a key challenge in real-world scenarios, where data schedules are often unknown and possibly dynamic. CL algorithms should carefully satisfy (5) for safe deployment.

We note that schedule-robustness is a more general and stronger notion than order-robustness from Yoon et al. (2020). Our definition applies to online task-free CL while order-robustness only considers offline task-based setting. We also allow arbitrary ordering for individual samples instead of task-level ordering. In the following, we will present our method and show how it achieves schedule-robustness.

## 3 METHOD

We present **SC**hedule-**R**obust **O**nline continua**L** **L**earning (SCROLL) as a two-stage process: 1) learning online a schedule-robust predictor for CL, followed by 2) adapting the predictor using only replay data. In the first stage, we consider two schedule-robust predictors and discuss how to initialize them, motivated by the meta-CL perspective introduced in (4). In the second stage, we tackle how to adapt the predictors from the first stage with ER and the buffering strategy. We will show that SCROLL satisfies schedule-robustness *by construction*, given that optimizing a CL algorithm against all possible schedules as formulated in (5) is clearly intractable.

### 3.1 SCHEDULE-INVARIANT ONLINE PREDICTOR

We model the predictors introduced in (1) as the composition $f = \phi \circ \psi$ of a feature extractor $\psi : \mathcal{X} \rightarrow \mathbb{R}^m$ and a classifier $\phi : \mathbb{R}^m \rightarrow \mathcal{Y}$. In line with recent meta-learning strategies (e.g. Bertinetto et al., 2019; Raghu et al., 2020), we keep $\psi$ fixed during our method's first stage while only adapting the classifier $\phi$ to learn from data streams. We will discuss how to learn $\psi$ in Sec. 3.2.

A key observation is that some choices of $\phi$, such as Nearest Centroid Classifier (NCC) (Salakhutdinov & Hinton, 2007) and Ridge Regression (Kailath et al., 2000), are schedule-invariant by design.

**Nearest Centroid Classifier (NCC).** NCC classifies a sample $x$ by comparing it to the learned *"prototypes"* $c_y$ for each class $X^y \triangleq \{x | (x, y) \in D\}$ in the dataset,

$$f(x) = \arg\min_y \|\psi(x) - c_y\|_2^2 \quad \text{where} \quad c_y = \frac{1}{n} \sum_{x \in X^y} \psi(x). \tag{7}$$

The prototypes $c_y$ can be learned online: given a new $(x, y)$, we update only the corresponding $c_y$ as

$$c_y^{\text{new}} = \frac{n_y \cdot c_y^{\text{old}} + \psi(x)}{n_y + 1}, \tag{8}$$

where $n_y$ is the number of observed samples for class $y$ so far. We note that the prototype for each class is invariant to any ordering of $D$ once the dataset has been fully observed. Since $\psi$ is fixed, the resulting predictor in (7) is schedule-invariant and online. Further, $f$ is unaffected by catastrophic forgetting by construction, as keeping $\psi$ fixed prevents forgetting while learning each prototype independently mitigates cross-class interference (Hou et al., 2019).

**Ridge Regression.** Ridge Regression enjoys the same desirable properties of NCC, being schedule-invariant and unaffected by forgetting (see Appendix A for a full derivation). Let OneHot$(y)$ denote

---

[2]A formal definition is possible but outside the scope of the current work (see Appendix E for more details).

the one-hot encoding of class $y$, we obtain the predictor $f$ as the one-vs-all classifier

$$f(x) = \arg\max_y \ w_y^\top \psi(x), \qquad \text{where}$$

$$W^* = [w_1 \dots w_K] = \arg\min_W \frac{1}{|D|} \sum_{(x,y)\in D} \left\| W^\top \psi(x) - \text{OneHot}(y) \right\|^2 + \lambda \|W\|^2 . \tag{9}$$

Ridge Regression admits online updates via recursive least squares (Kailath et al., 2000): given $(x,y)$,

$$w_z = (A^{\text{new}} + \lambda I)^{-1} c_z^{\text{new}} \qquad \forall z \in \mathcal{Y}, \qquad \text{where}$$

$$A^{\text{new}} = A^{\text{old}} + \psi(x)\psi(x)^\top \qquad \text{and} \qquad c_z^{new} = \begin{cases} c_z^{\text{old}} + \psi(x) & \text{if } z = y \\ c_z^{\text{old}} & \text{otherwise.} \end{cases} \tag{10}$$

Here, $A$ denotes the covariance matrix of all the samples observed so far during a schedule, while $c_z$ is the sum of the embeddings $\psi(x)$ of samples $x$ in class $z$.

## 3.2 PREDICTOR INITIALIZATION

Within our formulation, initializing $f_0$ consists of learning a suitable feature representation $\psi$. As discussed in Sec. 2, we could learn $f_0 = \phi_0 \circ \psi$ by solving the meta-CL problem in (4) (with $\phi_0$ a classifier for the meta-training data to be discarded after initialization, see Fig. 1 (Right)). As shown in Sec. 3.1, if we set $\text{Alg}(\cdot)$ in (4) to be the online versions of NCC or Ridge Regression, the resulting predictor $f_T$ becomes invariant to any schedule over $D$. This implies that *the meta-CL problem is equivalent to a standard meta-learning one where all data in $D$ are observed at once*, namely

$$f_0 = \arg\min_f \ \mathbb{E}_{D\sim\mathcal{T}} \ \mathbb{E}_{(x,y)\sim\pi_D} \ell(f_D(x), y) \qquad \text{where} \qquad f_D = \text{Alg}(D, f). \tag{11}$$

For example, setting $\text{Alg}(\cdot)$ to NCC recovers exactly the well-known ProtoNet (Snell et al., 2017).

**Meta-learning and pre-training.** Meta-learning methods for solving (11) are mostly designed for low data regimes with small $D$ while in contrast, CL has to tackle long sequences (i.e. $D$ is large). In these settings, directly optimizing (11) quickly becomes prohibitive since the cost of computing gradients scales linearly with the size of $D$. Alternatively, recent works in meta-learning strongly support learning $f_0$ via standard multi-class classification (Tian et al., 2020; Wang et al., 2021). In particular, Wang et al. (2021) showed that the cross-entropy loss in multi-class classification is an upper bound to meta-learning $f_0$ in (11). Classification thus offers a theoretically sound estimator for meta-learning. As a direct consequence, our meta-learning perspective on initializing $f_0$ provides a principled justification for exploiting pre-training in CL. Pre-training is computationally efficient, since gradients can be computed on small batches rather than the entire $D$ in (11). Extensive evidence also suggests that pre-training often yields more robust $f_0$ than meta-learning (El Baz et al., 2022).

**Learning $f_0$ in SCROLL.** Given the reasoning above, we propose to not tackle (11) directly but rather learn $f_0$ by multi-class classification. Following the standard practices from meta-learning, we ensure the meta-training data to be disjoint from the sequences we run the CL algorithm on, such that $\text{Alg}(\cdot)$ is indeed learning novel data during CL. For instance, in Sec. 4.1 we will use $f_0$ trained on Meta-Dataset (Triantafillou et al., 2019) to perform CL on CIFAR datasets, which do not overlap.

## 3.3 PREDICTOR ADAPTATION WITH EXPERIENCE REPLAY

The predictor $f_T$ obtained via online learning is schedule-invariant and readily available for classification. However, it shares the same feature extractor $\psi$ with $f_0$ and may therefore suffer from the distribution shift between the pre-training data and $D$. We thus propose to further adapt $f_T$ into a $f^*$ using Experience Replay (ER). Our key insight is to consider buffering strategies that yield similar replay buffers, regardless of the schedule over $D$, hence keeping $f^*$ schedule-robust.

**Buffering strategy.** We adopt the *exemplar buffering* from Rebuffi et al. (2017). The method uses greedy moment matching to select the same number of samples for each class independently, according to how well the mean embedding (via $\psi$) of the selected samples approximates the mean embedding of all points in the class. The greedy selection provides a total ordering for all points in each class, which is used to determine which samples to store or discard. This step filters out outliers and ensures that $M_T$ contains representative samples from each class.

This process aims to keep each $M_t$ balanced class-wise, hence the final $M_T$ attains a similar class distribution after $D$ is fully observed, irrespective of the specific schedule. In particular, it is easy to show that $M_T$ is *schedule-invariant* if the data schedule present one entire class at a time (namely no $X^y$ is split across multiple batches), since the replay exemplars selected for each class is deterministic for such schedules. More generally, adapting $f_T$ with $M_T$ yields a schedule-robust $f^*$. We will further discuss schedule-robustness with respect to smaller batch sizes in Appendix B.4. Below we consider two adaptation strategies.

**Small buffer size: residual adapters.** When $M$ has limited capacity (e.g., 200 or 500 in our experiments), there may be insufficient data for adapting all parameters in $f_T$. To avoid overfitting, we introduce residual adapters inspired by task-specific adapters (Li et al., 2022). Concretely, we insert residual layers (e.g. residual convolutions) to each level of $\psi$. Then, while keeping $\psi$ fixed, we jointly optimize the new layers and the classifier $\phi$ using a cross-entropy loss (see Fig. 1 (Right)).

**Large buffer size: model tuning.** For larger buffers, we empirically observed that updating all parameters of $f_T$ is more effective. This is not surprising given the added representational power from the full model. We will also demonstrate that the threshold for switching from residual adapters to full-model tuning appears consistent and may be determined a priori.

### 3.4 PRACTICAL ALGORITHM

We present the complete algorithm in Alg. 1 (see also Fig. 1): We first pre-train $\psi$ via multi-class classification using the cross-entropy loss (Sec. 3.2) and initialize $f_0$ with the pre-trained $\psi$. As we receive data online from the sequence $S(D)$, we update the replay buffer $M_t$ (Sec. 3.3) and the necessary statistics for computing the classifier $\phi_t$ (via e.g. NCC or Ridge). At each step $t$ we can output the intermediate model $f_t = \phi_t \circ \psi$ (Sec. 3.1). The final model $f^*$ is obtained by adapting $f_T$ using $M_T$ via cross-entropy loss (Sec. 3.3).

We highlight several properties of SCROLL: 1) it is schedule-robust since we have shown in Sec. 3.1 that $f_T$ is schedule-invariant, and in Sec. 3.3 that adapting it to $f^*$ is schedule-robust. Therefore our overall algorithm combining the two components is schedule-robust. 2) it is class-incremental with a single classifier for all observed classes; 3) it meets the definition of online CL (Chaudhry et al., 2019a; Mai et al., 2022), where the data stream is observed once, and only the replay data is reused.

---

**Algorithm 1** SCROLL

**Input:** Pre-trained $\psi$, CL sequence $S(D)$, Buffer $M_0 = \varnothing$, buffer max size $m$.

$f_T, M_T = \text{ONLINE-CL}(\psi, S(D), M_0)$
**If** $|M_T| \le m$:
    $f^* = \text{ResidualAdapt}(f_T, M_T)$
**Else:**
    $f^* = \text{FullAdapt}(f_T, M_T)$
**return** $f^*$

**def** ONLINE-CL $(\psi, S(D), M_0)$
  **For** $B_t$ in $S(D)$:
    Update $M_t$ (exemplar strategy, Sec. 3.3)
    Update statistics $c_y, A$ using (8) or (10).
  Compute the classifier $\phi_t$ via (7) or (10).
  **return** $f_T = \phi_T \circ \psi$ and $M_T$

---

**Memory-free SCROLL.** We note that the $f_T$ returned by the Online-CL routine in Alg. 1, is already a valid and deterministic predictor for $D$. Hence, if we have constraints (e.g., privacy concerns Shokri & Shmatikov (2015)) or memory limitations, SCROLL can be memory-free by not using ER. We empirically investigate such strategy in our ablation studies Sec. 4.3 and Appendix B.2.

## 4 EXPERIMENTS

We consider class-incremental learning and compare SCROLL primarily to online CL methods, including GDumb (Prabhu et al., 2020), MIR (Aljundi et al., 2019a), ER-ACE and ER-AML (Caccia et al., 2022), and SSIL (Ahn et al., 2021). As shown by (Buzzega et al., 2020), online methods often perform poorly due to the limited number of model updates. To disentangle forgetting from underfitting in the online setting, we further compare to recent offline methods for completeness, including ER (Riemer et al., 2019), BIC (Wu et al., 2019), DER and DER++ (Buzzega et al., 2020). We recall that the conventional offline setting allows multiple passes and shuffles over data for each task (De Lange et al., 2021). We differentiate our method by the classifier used, namely SCROLL (NCC) and SCROLL (Ridge).

**Setup.** We first consider sequential CIFAR-10 and sequential CIFAR-100 benchmarks. We use a ResNet18 initial predictor trained on Meta-Dataset using multi-class classification (Li et al., 2021).

| | CL Algorithm | CIFAR-10 5-split | | | CIFAR-100 10-split | | |
|---|---|---|---|---|---|---|---|
| | Joint Training (i.i.d.) | 94.2 ±1.6 | | | 79.3 ±0.4 | | |
| | **CL Algorithm** | $|M| = 200$ | $|M| = 500$ | $|M| = 2000$ | $|M| = 200$ | $|M| = 500$ | $|M| = 2000$ |
| OFFLINE | ER | 67.9 ±1.1 | 77.3 ±0.4 | 86.8 ±0.4 | 22.9 ±0.6 | 33.2 ±0.6 | 53.1 ±0.3 |
| | BIC | **77.4** ±2.2 | **88.6** ±5.3 | **90.4** ±0.5 | **34.9** ±1.6 | **45.2** ±1.9 | 57.5 ±0.7 |
| | DER | 69.7 ±1.7 | 78.2 ±0.8 | 86.6 ±0.7 | 26.2 ±0.8 | 38.1 ±2.1 | 55.3 ±0.6 |
| | DER++ | 76.8 ±1.7 | 81.6 ±2.2 | 86.8 ±0.8 | 28.5 ±0.7 | 42.7 ±1.1 | **59.1** ±0.5 |
| ONLINE | GDumb | 73.6 ±1.9 | 81.2 ±0.9 | 87.3 ±0.8 | 21.9 ±1.3 | 39.0 ±1.0 | 59.8 ±0.2 |
| | ER-ACE | 77.8 ±0.9 | 81.4 ±0.6 | 84.7 ±0.8 | 40.3 ±1.2 | 49.0 ±0.9 | 56.5 ±1.0 |
| | ER-AML | 69.9 ±2.9 | 75.3 ±0.7 | 80.6 ±0.6 | 32.4 ±0.5 | 41.9 ±1.0 | 51.0 ±1.2 |
| | SSIL | 68.9 ±1.4 | 70.8 ±1.4 | 73.2 ±1.7 | 46.6 ±1.0 | 53.9 ±0.6 | 59.2 ±0.5 |
| | MIR | 55.2 ±1.5 | 68.7 ±1.3 | 82.0 ±1.0 | 23.2 ±0.8 | 31.9 ±0.5 | 47.5 ±1.1 |
| (Ours) | SCROLL (NCC) $f_T$ | | 65.3 ±0 | | | 40.4 ±0 | |
| | SCROLL (NCC) $f_T \rightarrow f_*$ | 81.5 ±0.1 | 84.6 ±0.1 | 88.4 ±0.1 | 48.3 ±0.1 | 56.4 ±0.2 | 66.6 ±0.2 |
| | SCROLL (Ridge) $f_T$ | | 81.7 ±0 | | | 57.1 ±0 | |
| | SCROLL (Ridge) $f_T \rightarrow f_*$ | **84.0** ±0.2 | **86.3** ±0.2 | **89.4** ±0.2 | **59.9** ±0.1 | **61.5** ±0.1 | **68.0** ±0.2 |

**Table 1:** Class-incremental classification accuracy on sequential CIFAR-10/100. Joint training accuracy obtained by training on all classes with standard supervised learning. Best online/offline methods in bold.

Following the convention of meta-learning, Meta-Dataset has no class overlap with CIFAR datasets. This ensures that CL algorithms must learn about the new classes during CL. We also evaluate SCROLL on *mini*IMAGENET as another sequential learning task to highlight that even a weakly initialized $f_0$ is very beneficial. Following Javed & White (2019), all methods are allowed to learn an initial predictor using the meta-training set of *mini*IMAGENET (38400 samples over 64 classes), and perform CL on the meta-testing set (20 novel classes). A ResNet12 is used on *mini*IMAGENET.

**Baseline methods.** For fairness, *all baseline methods use the same pre-trained feature extractor*. We also performed extensive hyper-parameter tuning for all baseline methods such that they perform well with pre-training (see Appendix D). Lastly, we standardize that buffer size reported in all experiments as the total buffer for all classes, not buffer per class.

## 4.1 CLASS-INCREMENTAL LEARNING ON CIFAR DATASETS

For CIFAR datasets, we first follow a standard schedule from Buzzega et al. (2020); Wu et al. (2019): CIFAR-10 is divided into 5 splits of 2 classes each while CIFAR-100 into 10 splits of 10 classes each. At each split, this schedule abruptly changes from one data distribution to another.

Tab. 1 shows that SCROLL (Ridge) outperforms all baselines by large margins on CIFAR-100 for all buffer sizes. For CIFAR-10, SCROLL only lags behind BIC for buffer sizes 500 and 2000 while outperforming the rest. However, we emphasize that BIC is an offline method and not directly comparable to ours. In addition, we will demonstrate next that the baseline methods including BIC have unreliable performance when the schedule changes.

**Varying Schedules.** We now consider the effect of using schedules for CIFAR-100 with splits presenting 20/10/4/2 classes at a time (respectively 5/10/25/50-splits). Fig. 2, evaluates the best performing baselines from Tab. 1 under these schedules. We observe that SCROLL performs consistently across all settings and outperforms all baselines by a large margin, with BIC performing competitively only under the 5-split. Additionally, we clearly see that the accuracy of both online and offline methods drops drastically as the dataset split increases. This further validates Yoon et al. (2020); Mundt et al. (2022) who observed that existing methods are brittle to schedule variations. The only exception is GDumb, which only uses replay data and is thus (mostly) robust against schedule changes, according to our discussion in Sec. 3.3. We note, however, that GDumb performs poorly in some "worst-case" regimes (see Appendix B.3). Additional results are reported in Appendices B.1 and B.2. including a comparison with (Shanahan et al., 2021) who introduces an ensemble CL method to tackle schedules variations, and a challenging Gaussian schedule for empirical evaluation.

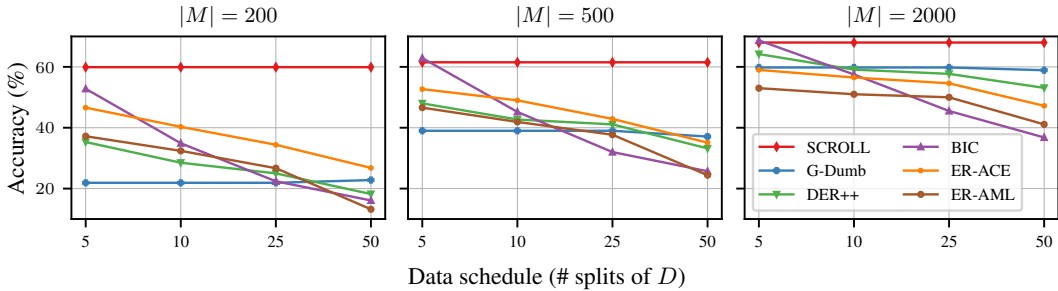

**Figure 2:** Robustness of SCROLL (Ridge) vs offline/online baselines to different schedules of CIFAR-100.

## 4.2 CLASS-INCREMENTAL LEARNING ON *mini*IMAGENET

In Tab. 2, qe evaluate SCROLL on *mini*IMAGENET 10-split following the setup described earlier. We stress that all methods perform pre-training on the small meta-training set and then carry out CL on the meta-test set. We note that SCROLL (Ridge) outperforms all baseline methods with noticeable improvements. This suggests that our method does not require pre-training on massive datasets and could work well with limited data. Further, Ridge Regression classifier clearly outperforms NCC classifier. We further investigate this phenomenon in the ablation studies.

| | **CL Algorithm** | $|M| = 200$ | $|M| = 500$ |
|---|---|---|---|
| | *mini*IMAGENET 10-split | | |
| | Joint Training (i.i.d.) | 89.2 $\pm$0.4 | |
| OFFLINE | BIC | 64.6 $\pm$3.3 | 73.4 $\pm$1.6 |
| | ER | 63.7 $\pm$1.0 | 73.6 $\pm$0.6 |
| | DER | 67.3 $\pm$1.1 | 75.2 $\pm$0.8 |
| | DER++ | 68.8 $\pm$1.0 | 78.0 $\pm$0.6 |
| ONLINE | GDumb | 65.6 $\pm$1.3 | 74.5 $\pm$0.5 |
| | ER-ACE | 57.5 $\pm$1.4 | 64.4 $\pm$0.6 |
| | ER-AML | 52.4 $\pm$2.2 | 58.4 $\pm$1.7 |
| | SSIL | 57.0 $\pm$1.5 | 63.3 $\pm$1.6 |
| | MIR | 48.4 $\pm$2.8 | 67.3 $\pm$2.9 |
| (ours) | SCROLL (NCC) | 72.8 $\pm$0.2 | 76.0 $\pm$0.1 |
| | SCROLL (Ridge) | **80.9** $\pm$0.1 | **82.0** $\pm$0.1 |

**Table 2:** Classification accuracy on sequential *mini*IMAGENET 10-split.

## 4.3 ABLATION STUDIES

**Effects of classifier initialization.** In our formulation of ER, the role of NCC or Ridge Regression is to initialize the classifier $\phi$ at $f_T$ before adapting $f_T$ towards $f^*$. In principle, one could skip this online learning step (i.e. Online-CL routine in Alg. 1), initialize $\phi$ in $f_T$ randomly, and adapt $f_T$. Tab. 3 compares these options for initializing the classifier[3].

The results suggest that how classifier $\phi$ is initialized at $f_T$ drastically affect the performance of final predictor $f^*$. In particular, the performance of $f_T$ correlates strongly with the performance of $f^*$: Ridge Regression has the best predictor both before and after adaptation. In contrast, randomly initialized $\phi$ does not learn from data streams, and the resulting $f^*$ falls short of the other two initializers. Crucially, the

| $\phi$ **Init** | CIFAR-100 10-split | | *mini*IMAGENET 10-split | |
|---|---|---|---|---|
| | $f_T$ | $\rightarrow$ $f^*$ | $f_T$ | $\rightarrow$ $f^*$ |
| Random | 1.2 $\pm$0.3 | 49.0 $\pm$0.2 | 4.9 $\pm$0.3 | 75.3 $\pm$0.8 |
| NCC | 40.4 $\pm$0 | 56.4 $\pm$0.2 | 65.3 $\pm$0 | 76.0 $\pm$0.1 |
| Ridge | 57.1 $\pm$0 | 61.5 $\pm$0.1 | 80.6 $\pm$0.2 | 82.0 $\pm$0.1 |

**Table 3:** Classification accuracy with different classifier initialization for $|M| = 500$.

results suggest that online learning described in Sec. 3.1 is vital for fully exploiting the pre-training. We also remark that while randomly initialized $\phi$ resembles GDumb, there are still crucial differences between them, including buffering strategy, model architecture and training routine. In particular, we study in depth the role of the buffering strategy in Appendix B.3.

**Effects of replay buffer size.** We study the impact of changing replay buffer capacity for SCROLL. We focus on SCROLL (Ridge), given its better performance over the NCC variant. Tab. 4 reports how buffer capacity affect model performance using either residual adapters or full-model tuning.

We first observe that ER contributes significantly to final performance. Using Ridge Regression, the memory-free $f_T$ only obtains 57% for CIFAR-100 using Ridge Regression. Even with a limited buffer of 200 (i.e. 2 samples per class), ER adds about 3% to the final accuracy. When $M = 2000$, ER adds more than 10%. The results suggest that updating the pre-trained representation and the associated classifier is important. When combined with the effects from classifier initialization, the results suggest that all components in our method are important for learning robust predictors.

---

[3]We will discuss in Appendix C how to implement (7) as a standard linear layer

| | CIFAR-10 | | | CIFAR-100 | | |
|---|---|---|---|---|---|---|
| $f_T$ Accuracy | $81.7 \pm 0$ | | | $57.1 \pm 0$ | | |
| **Adaptation Routine** | $\|M\| = 200$ | $\|M\| = 500$ | $\|M\| = 2000$ | $\|M\| = 200$ | $\|M\| = 500$ | $\|M\| = 2000$ |
| Residual Adapter | $\mathbf{84 \pm 0.2}$ | $\mathbf{86.3 \pm 0.2}$ | $89.0 \pm 0.1$ | $\mathbf{59.9 \pm 0.1}$ | $\mathbf{61.5 \pm 0.1}$ | $64.7 \pm 0.2$ |
| Full-model tuning | $81.5 \pm 0.03$ | $84.1 \pm 0.4$ | $\mathbf{89.4 \pm 0.2}$ | $56.5 \pm 0.3$ | $61.1 \pm 0.2$ | $\mathbf{68.0 \pm 0.2}$ |

**Table 4:** Classification accuracy of different adaptation routines for SCROLL (Ridge).

Tab. 4 also suggests that full-model tuning causes overfitting when replay data is limited. At $|M| = 200$, the final predictor $f^*$ consistently underperforms $f_T$. As the buffer size increases, full-model tuning becomes more effective compared to residual adapters. For CIFAR-100, full-model tuning is over 3% higher at $|M| = 2000$. We also stress that the residual adapters are still usable with high buffer capacity, improving nearly 8% from $f_T$. For both datasets, we set the threshold $m = 500$ for the practical algorithm to switch from residual adapters to full-model tuning.

## 5 DISCUSSION

In this work, we introduced the notion of *schedule-robustness* in CL, which requires algorithms to perform consistently across different data streams. We argued that such property is key to CL applications, where the schedule is often unknown a priori. However, most existing methods disregard this aspect and are designed around specific schedules. We empirically demonstrated that this severely degrades their performance when a different schedule is used. To tackle these issues, we proposed SCROLL and formulated CL as learning online a schedule-robust classifier, followed by predictor adaptation using only replay data. SCROLL is schedule-robust by design and outperforms existing methods across all evaluated schedules. We now focus on two key aspects that emerged from this work, which we believe provide useful insights for CL:

**Pre-training and CL.** The results from our experiments show that pre-training is broadly beneficial for all evaluated methods (see our Tab. 1 vs Tab. 2 in Buzzega et al. (2020)). We believe that this observation should call for *greater focus on how to exploit pre-training to develop robust CL methods*: our choice to pre-train the feature extractor $\psi$ in SCROLL and combining it with linear online methods such as incremental Ridge Regression, is a principled decision derived from our meta-learning-based perspective, which also helps to mitigate forgetting. In contrast, most previous CL methods adopting pre-training see it as a preliminary phase rather than an integral component of their methods (Wang et al., 2022; Shanahan et al., 2021; Wu et al., 2022). This approach often leads to sub-optimal design decisions and potentially less effective techniques for leveraging pre-training as previously observed in Mehta et al. (2021) and further highlighted by our experiments (see also Appendix B.2 where SCROLL (Ridge) outperforms the more sophisticated ensemble model Shanahan et al. (2021)).

Pre-training is also a realistic assumption. We have argued that application domains (e.g., computer vision and natural language understanding) have vast amounts of curated data and increasingly available pre-trained models that CL algorithms could readily exploit. Additionally, we demonstrated in Sec. 4.2 that pre-training with even a modest dataset is very beneficial, which is a promising result for applying this strategy to other domains. In particular, we note these settings are analogous to initializing a CL algorithm with a random model but then having a large first batch (i.e., containing samples from several classes), a schedule that is gaining increasing traction in the literature (Hou et al., 2019; Mittal et al., 2021). Our strategy extends naturally to these settings if we allow offline training on the first batch to replace pre-training.

**Class-independent Online Learning and On-demand Adaptation.** We believe that SCROLL provides a new general practice to approach CL problems. In particular, the strategy of *first learning online a coarse-yet-robust model for each individual class and then adapting it on-demand on data of interest* grants flexibility as well as consistency to CL algorithms. The online step focuses on learning each class independently, which prevents inter-class interference (Hou et al., 2019) by construction. This single-class learning process is open-ended (i.e. one-vs-all for each class) with any new classes easily assimilated. We have demonstrated that the predictor $f_T$ resulting from this first step, is already competitive with most previous CL algorithms. Then, at any moment $T$ in time (not necessarily at the end of the sequence), we can adapt $f_T$ to a $f^*$ specializing to all the observed classes so far. This step is performed with a cross-entropy loss, which enforces interaction among classes and ultimately leads to improved performance. SCROLL's robust performance offer strong support to adopting this strategy more broadly in CL.

**Limitation and Future Works.** We close by discussing some limitations of our work and future research directions. Firstly, while model adaptation works well in practice, most previously observed data are omitted from directly updating the feature extractor. One key direction we plan to investigate is how to perform the periodic update of the feature extractor *during CL*, to improve data efficiency while preserving schedule-robustness. Secondly, we note that the residual adapters in our model are specific to convolutional networks. We will extend our method to more powerful architectures like transformers.

**Reproducibility Statement.** As per ICLR 2023 guidelines[4], we will make sample code available to AC and reviewers via private message on OpenReview.net once the discussion forum will be opened. The code reproduces our results on CIFAR-100 for $|M| = 500$ and $|M| = 2000$, and on *mini*IMAGENET for $|M| = 500$. See README in the code for the setup guide.

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

# APPENDIX

This appendix is organized in the following manner:

- In Appendix A we comment on the schedule-robustness property of Ridge Regression;
- In Appendix B we present additional empirical investigation on the behavior of SCROLL and its main components;
- In Appendix C we report the implementation details of SCROLL used in our experiments;
- In Appendix D we provide more information on model tuning and baseline methods.
- In Appendix E we discuss in more detail the definition of schedule-robustness introduced in Sec. 2.2 providing a more formal definition of this notion.

## A  RIDGE REGRESSION AND SCHEDULE-ROBUSTNESS

We show that Ridge Regression is schedule-robust and mitigates forgetting. From Sec. 3.1, the Ridge Regression classifier is defined as $f(x) = \arg\max_y w_y^\top \psi(x)$, where

$$W^* = [w_1 \dots w_K] = \arg\min_W \frac{1}{|D|} \sum_{(x,y) \in D} \left\| W^\top \psi(x) - \text{OneHot}(y) \right\|^2 + \lambda \|W\|^2, \qquad (12)$$

with $\psi$ as a feature extractor. The closed-form solution for $W$ is

$$W^* = (X^\top X + \lambda I)^{-1} X^\top Y \qquad \text{where} \qquad (13)$$

$$X = [\psi(x_1) \dots \psi(x_N)]^\top \text{ and } Y = [\text{OneHot}(y_1) \dots \text{OneHot}(y_N)]^\top. \qquad (14)$$

Substituting (14) into (13), we have

$$X^\top X = \sum_{i=1}^N \psi(x_i)\psi(x_i)^\top \text{ and } X^\top Y = [c_1 \dots c_K], \qquad (15)$$

where $c_i = \sum_{x \in X^i} \psi(x)$ is the sum of embeddings from class $X^i$, and $K$ is the total number of classes in the dataset $D$.

From (15), it is clear that both $X^\top X$ and $X^\top Y$, as summations, are invariant to the ordering of samples in $D$. Therefore computing $W^*$ is also invariant to sample ordering in $D$ and may be performed online using (10). We thus conclude that Ridge Regression is schedule-invariant. Since Ridge Regression learns each $w_i$ independently, it also mitigates catastrophic forgetting caused by class interference.

We note that while Ridge regression with a fixed feature representation resembles streaming linear discriminant analysis with deep models Hayes & Kanan (2020), a key difference is that the former guarantees schedule-invariance while the latter is sensitive to sample ordering..

## B  ADDITIONAL EXPERIMENTS

### B.1  ADDITIONAL RESULTS ON DIFFERENT SPLITS FOR CIFAR DATASETS

We compare SCROLL with the baseline methods on CIFAR-10 10-split and CIFAR-100 50-split respectively. The classification accuracies are reported in Tab. 5.

SCROLL (Ridge) outperforms all baselines in Tab. 5. We observe that most baseline methods' performance drop drastically compared to Tab. 1 except GDumb, suggesting worse forgetting with more dynamic schedules, even though the underlying data remains unchanged. In contrast, SCROLL is able to perform consistently due to its schedule-robust design. The drop in baselines performance is also consistent with Boschini et al. (2022): most existing methods are not designed to effectively exploit pre-training. When the same dataset is split into more tasks (or batches), the later ones benefit less from the prior knowledge encoded in the initial predictor $f_0$.

| | | CIFAR-10 10-split | | | CIFAR-100 50-split | | |
|---|---|---|---|---|---|---|---|
| Joint Training (i.i.d.) | | 94.2 ±1.6 | | | 79.3 ±0.4 | | |
| **CL Algorithm** | | $\|M\| = 200$ | $\|M\| = 500$ | $\|M\| = 2000$ | $\|M\| = 200$ | $\|M\| = 500$ | $\|M\| = 2000$ |
| OFFLINE | ER | 54.8 ±1.8 | 71.5 ±1.1 | 85.8 ±0.1 | 15.1 ±2.1 | 28.5 ±1.1 | 50.3 ±0.4 |
| | BIC | 64.9 ±2.0 | 78.8 ±0.5 | 87.5 ±0.6 | 16.1 ±4.1 | 25.7 ±0.6 | 36.8 ±2.7 |
| | DER | 54.4 ±5.2 | 63.2 ±2.1 | 75.4 ±7.4 | 15.7 ±1.7 | 29.6 ±2.0 | 48.2 ±1.4 |
| | DER++ | 62.9 ±3.1 | 68.8 ±2.6 | 77.4 ±5.8 | 18.2 ±1.4 | 33.2 ±2.0 | 53.1 ±0.9 |
| ONLINE | GDumb | 73.0 ±1.7 | 81.1 ±0.3 | 87.1 ±0.1 | 22.8 ±1.9 | 37.1 ±1.1 | 58.9 ±0.4 |
| | ER-ACE | 69.8 ±1.8 | 77.3 ±0.8 | 81.7 ±1.2 | 26.8 ±0.9 | 35.1 ±0.9 | 47.2 ±0.5 |
| | ER-AML | 72.8 ±1.7 | 80.0 ±0.3 | 85.2 ±0.6 | 13.2 ±1.1 | 24.4 ±1.8 | 41.1 ±1.1 |
| | SSIL | 65.1 ±1.6 | 70.7 ±0.4 | 74.2 ±1.0 | 32.9 ±0.6 | 40.1 ±0.9 | 52.2 ±0.9 |
| | MIR | 52.6 ±2.5 | 63.8 ±1.5 | 72.5 ±1.7 | 7.9 ±0.7 | 17.1 ±0.7 | 41.1 ±0.5 |
| (ours) | SCROLL (NCC) | 81.5 ±0.1 | 84.6 ±0.1 | 88.4 ±0.1 | 48.3 ±0.1 | 56.4 ±0.2 | 66.6 ±0.2 |
| | SCROLL (Ridge) | **84** ±0.2 | **86.3** ±0.2 | **89.4** ±0.2 | **59.9** ±0.1 | **61.5** ±0.1 | **68.0** ±0.2 |

**Table 5:** Class-incremental classification accuracy on CIFAR-100 50-split and CIFAR-10 10-split.

## B.2 COMPARISON WITH ENSEMBLE METHOD

We also compare SCROLL to ENS (Shanahan et al., 2021), which is designed to tackle different schedules, including a Gaussian schedule introduced in the same paper (see Fig. 3). Specifically, ENS keeps the pre-trained feature extractor fixed and learns an ensemble of classifiers for the novel classes. For ENS, we keep its original design for pre-training: using ResNet50 via self-supervised learning on ImageNet [5]. We also include the memory-free variant of SCROLL (Ridge) in the comparison, since ENS does not use ER.

| | CIFAR-10 | | | CIFAR-100 | | |
|---|---|---|---|---|---|---|
| **CL Algorithms** | **5-split** | **10-split** | **Gaussian schd** | **20-split** | **100-split** | **Gaussian Schd** |
| ENS (ResNet50) | 79.0 ±0.4 | 78.3 ±0.4 | 50.1 ±9.5 | 55.3 ±0.4 | 54.1 ±0.5 | 39.0 ±1.4 |
| SCROLL (ResNet50, $\|M\| = 0$) | | **88.1** ±0 | | | **67.2** ±0 | |
| SCROLL (ResNet18, $\|M\| = 0$) | | 81.7 ±0 | | | 57.1 ±0 | |
| SCROLL (Resnet18, $\|M\| = 500$) | | 86.3 ±0.2 | | | 61.5 ±0.1 | |

**Table 6:** Classification accuracy comparison SCROLL (Ridge) vs. Ensemble.

In Tab. 6, we observe that the ResNet50 provided by ENS is substantially more powerful than the ResNet18 used in our main experiments. In addition, memory-free SCROLL returns deterministic predictors $f_T$ for all possible schedules, while the performance of ENS degrades on the Gaussian schedule, despite its robustness in handling different splits. The results further validates that it is intractable to optimize CL algorithms against all schedules, and schedule-robustness should be considered as part of algorithm design. Lastly, we note that SCROLL has the additional advantage of using ER to further improve performance, compared to the lack of ER in ENS.

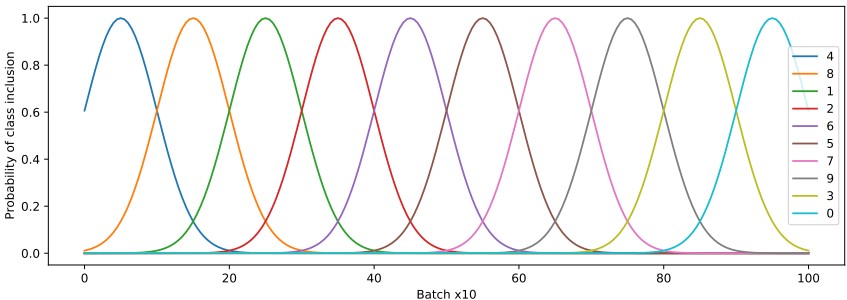

**Figure 3:** Example of Gaussian schedule. Figure originally from Shanahan et al. (2021). The schedule proposes to sample a class with a Gaussian probability, where each class peaks at a different time. There are no class boundaries, and the probability of sampling each class evolves smoothly over time.

---

[5]ResNet50 checkpoint available at https://github.com/deepmind/deepmind-research/tree/master/byol

## B.3 THE EFFECTS OF BUFFERING STRATEGY

In this section, we study different buffering strategies used in ER. We consider both exemplar selection (Rebuffi et al., 2017) and random reservoir sampling (Vitter, 1985), which are commonly used in existing works. We also introduce two simple variants to exemplar selection, including selecting samples nearest to the mean representation of each class (Nearest Selection), or selecting samples furthest to the mean representation of each class (Outlier Selection).

| Buffering strategy | CIFAR-10 | | CIFAR-100 | |
|---|---|---|---|---|
| | $|M| = 500$ | $|M| = 2000$ | $|M| = 500$ | $|M| = 2000$ |
| Exemplar | 86.3 $\pm 0.2$ | 89.4 $\pm 0.2$ | 61.5 $\pm 0.1$ | 68.0 $\pm 0.2$ |
| Random | 84.5 $\pm 0.2$ | 88.8 $\pm 0.1$ | 61.1 $\pm 0.1$ | 65.4 $\pm 0.1$ |
| Nearest | 72.3 $\pm 0.1$ | 82.2 $\pm 0.1$ | 55.6 $\pm 0.1$ | 57.0 $\pm 0.2$ |
| Outlier | 72.6 $\pm 0.4$ | 82.1 $\pm 0.3$ | 55.5 $\pm 0.1$ | 57.0 $\pm 0.3$ |

**Table 7:** Classification accuracy with different buffering strategy

We report the effects of different buffering strategies in Tab. 7. The exemplar strategy clearly performs the best, followed by random reservoir sampling. Both nearest and outlier selection perform drastically worse. From the results, we hypothesize that an ideal buffering strategy should store representative samples of each class (e.g., still tracking the mean representation of each class as in exemplar selection), while maintaining sufficient sample diversity (e.g., avoid nearest selection). We remark that the results in Tab. 7 are obtained using a fixed feature extractor $\psi$. We leave it to future work to investigate the setting where $\psi$ is continuously updated during CL.

The results have several implications for CL algorithms. While random reservoir sampling performs well *on average*, it can perform poorly in the *worst cases*, such as when the random sampling happens to coincide with either nearest or outlier selection. Therefore, it appears helpful to rank the samples observed during CL and maintain representative yet diverse samples. On the other hand, the results show that GDumb (Prabhu et al., 2020) is not fully schedule-robust, since random reservoir sampling would perform poorly in worst cases.

## B.4 BUFFERING STRATEGY AND SCHEDULE-ROBUSTNESS

As discussed in Sec. 3.3, exemplar selection yields a deterministic replay buffer when one or more classes are presented in a single batch. In this section, we further investigate its behaviors under schedules with smaller batches, where data from a class is spread out in a sequence. We use random reservoir sampling as a baseline for comparison.

In this experiment, we consider each class separately, since both exemplar strategy and random sampling select samples from each class independently. We thus use buffer size $b_1$ and batch size $b_2$ on a per-class basis here. Batch size refers to the number of samples in a batch for a single class, while buffer size refers to the allowed buffer capacity per class.

We use the $\ell_2$ distance $\left\| \hat{\mu}_y - \mu_y^* \right\|$ as the metric to measure how well the stored samples in the last replay buffer $M_T$ approximates the population mean of each class, where $\hat{\mu}_y$ is the mean of the points in $M_T$ belonging to class $y \in \mathcal{Y}$ and $\mu_y^*$ the mean of $X^y$, namely all points belonging to class $y$ in the dataset $D$.

We consider 4 scenarios: 1) $b_1 = 20, b_2 = 20$; 2) $b_1 = 20, b_2 = 80$; 3) $b_1 = 90, b_2 = 10$; 4) $b_1 = 50, b_2 = 50$. We also note that $b_1 = 20$ represents one of the worst possible scenario for CIFAR-10, since it implies a total buffer size of 200 and no class is allowed to temporarily borrow capacity from other classes. All scenarios are run with 100 random shuffling of the class data to simulate various schedules.

Fig. 4 shows that the exemplar strategy is much better at approximating the population mean of each class for all scenarios, compared to random sampling. This corroborates our results from Appendix B.3 that exemplar selection selects more representative samples and thus performs better during ER. In addition, we note that exemplar selection achieves drastically lower variance for all settings, suggesting that the selected samples at $M_T$ form similar distributions and make SCROLL schedule-robust.

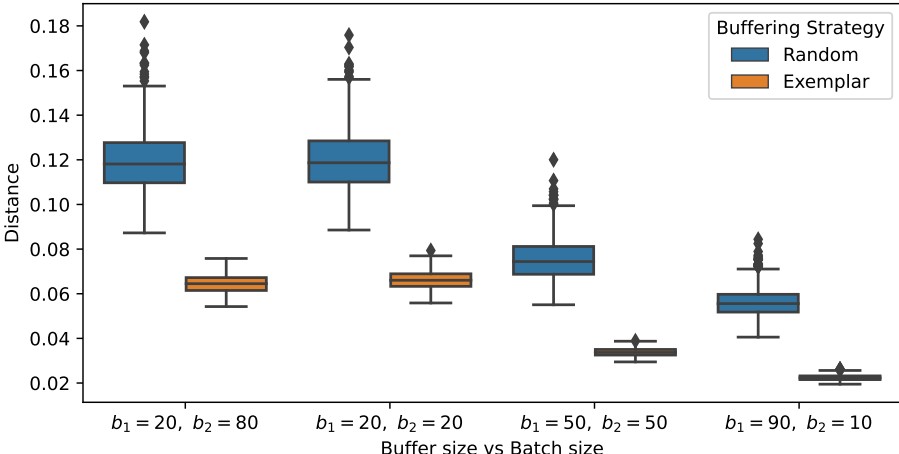

**Figure 4:** Effects of buffer size and batch size on selected samples in $M_T$. Exemplar selection vs. Random Sampling. Lower distance and variance are better. Distance denotes how well the selected samples approximates the population mean. Lower variance suggests similar distribution at $M_T$.

The experiment also suggests practical ways to ensure schedule-robustness. We first observe that the exemplar strategy decide on what samples to keep based on the current batch and the current buffer $M_t$. Therefore its behavior is predominantly determined by the combined size of $b_1$ and $b_2$. Consequently, simply use a large buffer (e.g. $b_1 = 90$ in scenario 4) will yield a $M_T$ closely approximating the desired population mean, even when the batch size $b_2 = 10$ is very small and generally considered as challenging for CL.

### B.5 GENERALIZATION PERFORMANCE OF INTERMEDIATE PREDICTORS

Existing CL methods can produce intermediate predictors for partial sequences, such as after observing each class. SCROLL can similarly do so *on demand*: whenever a predictor is required, SCROLL could compute the current predictor $f_t = \phi_t \cdot \psi$ with the stored statistics, followed by adapting $f_t$ on the current buffer $M_t$. The key difference from existing methods is that we will *always begin the adaptation from the initial $\psi$ to preserve schedule-robustness*.

In Fig. 5, we compare SCROLL (Ridge) to existing baselines with respect to the generalization performance of intermediate predictors. We note that the improved generalization performance directly implies reduced forgetting. The results show that SCROLL achieves performance similar to the baselines when the number of observed classes are low. However, SCROLL clearly outperforms the baselines with increasing number of observed classes. This clearly indicates that the proposed method is more robust in mitigating forgetting and achieves better generalization performance.

## C MODEL DETAILS

For CIFAR datasets, we use a ResNet18 backbone pre-trained on the Meta-Dataset (Triantafillou et al., 2019). The model definition and pre-trained representation can be found here[6]. For *mini*IMAGENET, we use a ResNet-12 backbone as commonly adopted in existing meta-learning works (Lee et al., 2019; Oreshkin et al., 2018; Ravichandran et al., 2019; Tian et al., 2020), and use the default architecture from the official implementation of Tian et al. (2020). Please refer to the accompanying source code for more details.

Following Tian et al. (2020), we find it beneficial to normalize the embedding $\psi(x)$ to unit length before classification. The normalization is implemented in predictor $f_t$ as a non-linearity layer.

---

[6] https://github.com/VICO-UoE/URL

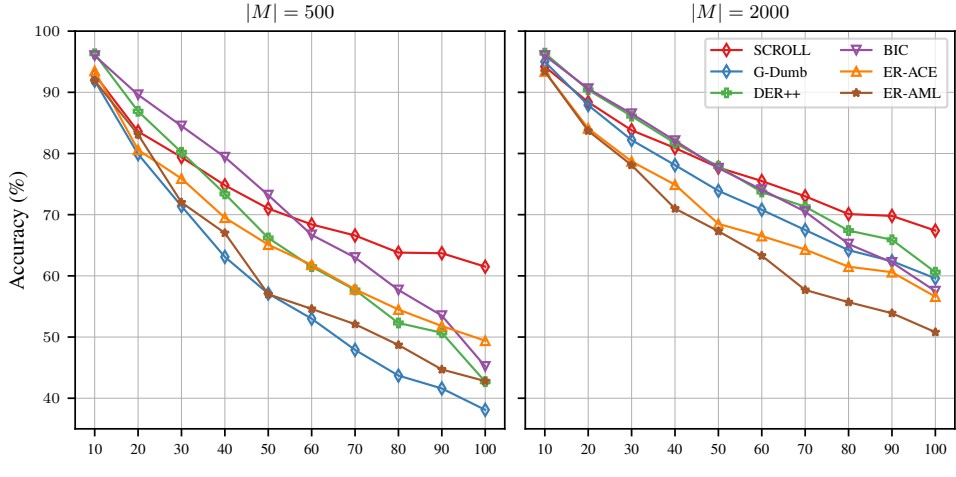

**Figure 5:** Intermediate predictors' performance on CIFAR-100 10-split schedule: SCROLL (Ridge) vs baselines.

**Adapting NCC with Experience Replay.** Since we normalize sample embedding, NCC can be implemented as a standard linear layer. Recall that NCC is

$$f(x) = \arg\min_y \|\psi(x) - c_y\|_2^2 \quad \text{s.t.} \quad c_y = \frac{1}{n} \sum_{x \in X^y} \psi(x). \tag{16}$$

When $\psi(x)$ is unit-length,

$$\arg\min_y \|\psi(x) - c_y\|_2^2 = \arg\min_y \ \psi(x)^\top \psi(x) - \psi(x)^\top c_y + c_y^\top c_y \tag{17}$$

$$= \arg\min_y \ 1 - \psi(x)^\top c_y + c_y^\top c_y \tag{18}$$

$$= \arg\max_y \ \psi(x)^\top c_y - c_y^\top c_y. \tag{19}$$

We thus use $c_y$ and $-c_y^\top c_y$ to initialize the weights and bias for class $y$ respectively. Adapting NCC with ER therefore treats NCC as a standard linear layer as described above.

**Optimizer.** Following Li et al. (2022), we use AdaDelta (Zeiler, 2012) for model adaptation with ER.

**ER Hyper-parameters.** We list the hyper-parameters for ER in Tab. 8 and state the values used in our grid search.

| Hyper-parameter | Description | Possible values |
|---|---|---|
| $l_1$ | learning rate for classifier $\phi$ | [0.5, 0.1, 0.01, 0.001] |
| $l_2$ | learning rate for feature extractor $\psi$ | [0.02, 0.01, 0.002, 0.001] |
| $\tau$ | temperature for cross-entropy loss | [1, 2, 4, 5] |
| $b$ | batch size from ER | [50, 100] |

**Table 8:** ER Hyper-parameters in the proposed method and their values used in the grid search.

## D   MODEL TUNING FOR BASELINE METHODS

### D.1   CIFAR-10 / 100

**Dataset.** We resize the input images to $84 \times 84$, both at training and test time, to be compatible with the pre-trained representation from Li et al. (2021). During training, we apply random cropping and horizontal flipping as standard data augmentations.

**Hyperparameters.** Starting from the exhaustive search in Buzzega et al. (2020), we tuned all the baselines by testing different learning rates and method-specific hyperparameters using the official

repository[7]. We used a vanilla SGD optimizer without momentum and batch size of 32, training for 50 epochs per task for all offline algorithms. We found that a good trade-off between preserving and updating the knowledge in the pre-trained representation is given by learning rates of $0.01$ and $0.03$. For small buffer sizes, typically $lr = 0.01$ achieves a lower standard deviation across different random seeds at the cost of slightly lower average performance, while $lr = 0.03$ works well with larger memory. Other empirical works also recommend updating the model parameters with lower learning rates by studying the training dynamics of CL algorithms (Mirzadeh et al., 2020). For **ER** and **BIC** Wu et al. (2019), we used $lr = 0.01$ (no other hyperparameters are present). For **DER/DER++** (Buzzega et al., 2020), we tested different values of replay loss coefficients $\alpha, \beta \in \{0.1, 0.3, 0.5, 1.0\}$, defined in Eq.6 of the original paper. While for small buffer sizes (200, 500), no clear winner emerges, we found that larger buffer sizes (2000) benefit from larger coefficients. Assuming that the actual data schedule is not known in advance, safe choices for $\alpha$ are $0.5$ or $0.3$, confirming the original paper suggestion. DER++ usually requires $\beta \geq 0.5$ and is less sensitive to the choice of $\alpha$. For CIFAR-100 10-split we used $lr = 0.01$, $\alpha = 0.5$, $\beta = 0.5$, while for the 50-split $lr = 0.03$ works best.

All online baselines use a single epoch for training. For **GDumb** (Prabhu et al., 2020), we decay $lr$ to $5 \times 10^{-4}$ with Cosine annealing and modify the original training recipe by removing CutMix regularization (Yun et al., 2019) as we found it not to be compatible with the pre-trained representation. For other online methods, we follow the official implementation[8] of **ER-AML** and **ER-ACE** (Caccia et al., 2022) and tunes the learning rate of each method. We observe similarly that $lr = 0.01$ works well for all. Loss coefficients are set using the recommended values in the original implementation.

Moreover, our experiments confirm that, if an algorithm is not schedule-robust, its hyper-parameters should be tuned independently for each schedule. However, this is infeasible as the actual schedule is generally unknown, further motivating the necessity of designing schedule-robust CL algorithms.

### D.2 *mini*IMAGENET

**Dataset.** For *mini*IMAGENET, we used the 64 classes of the meta-training split for pre-training the representation. Then, we train and test all CL algorithms on the 20 meta-test classes, by splitting the dataset in 10 sequential binary tasks. During training, we apply random cropping and horizontal flipping as standard data augmentations. Input images have shape $84 \times 84 \times 3$.

**Hyperparameters.** Similarly as for CIFAR-10/100, we search for the best hyperparameters of the baseline methods. We train all algorithms with SGD optimizer without momentum and batch size of 32. We found that for buffer size $|M| = 200$, training for 10 epochs is enough, while for $|M| = 500$, better performance are obtained with 20 epochs. Best hyperparameters are reported below.

*Buffer size $|M| = 200$ (10 epochs)*

- **ER**: $lr = 0.01$
- **BIC**: $lr = 0.01$
- **DER**: $lr = 0.03$, $\alpha = 1.0$
- **DER++**: $lr = 0.03$, $\alpha = 1$, $\beta = 1.0$

*Buffer size $|M| = 500$ (20 epochs)*

- **ER**: $lr = 0.01$
- **BIC**: $lr = 0.01$
- **DER**: $lr = 0.05$, $\alpha = 1.0$
- **DER++**: $lr = 0.05$, $\alpha = 0.5$, $\beta = 1.0$

*Buffer size $|M| = 200$ and $|M| = 500$ (1 epochs, online methods)*

- **GDumb**: $lr = 0.03$ decayed to $5 \times 10^{-4}$ with Cosine annealing, ER epoch $E = 10$, No Cutmix.

---

[7]https://github.com/aimagelab/mammoth
[8]https://github.com/pclucas14/aml

- **ER-ACE**: $lr = 0.01$
- **ER-AML**: $lr = 0.01$, supercon temperature $\tau = 0.2$
- **SSIL**: $lr = 0.01$, distillation coefficient $\alpha = 1$
- **MIR**: $lr = 0.01$, sub-sampling rate $r = 50$.

*Buffer size* $|M| = 500$   (1 epochs, online methods)

- **GDumb**: $lr = 0.03$ decayed to $5 \times 10^{-4}$ with Cosine annealing, ER epoch $E = 20$. No Cutmix.
- **ER-ACE**: $lr = 0.01$
- **ER-AML**: $lr = 0.01$, supercon temperature $\tau = 0.2$
- **SSIL**: $lr = 0.01$, distillation coefficient $\alpha = 1$
- **MIR**: $lr = 0.01$, sub-sampling rate $r = 50$.

# E    FORMAL DEFINITION OF SCHEDULE-ROBUSTNESS

In Sec. 2.2, we introduced the notions of schedule invariance and robustness. In particular, we said that a CL algorithm is **schedule-robust** if its performance across different schedules does not vary dramatically. We captured this intuition in (6) as

$$\mathcal{L}(S_1(D), f_0, \text{alg}) \approx \mathcal{L}(S_2(D), f_0, \text{Alg}), \qquad \forall S_1, S_2 \text{ schedules}, \ \forall D \text{ dataset},$$

where the symbol $\approx$ loosely implies that small discrepancies between generalization errors are allowed. This definition is however not formal, but rather conveying the general intuition and goal of this work, which is to design CL algorithms that are as robust as possible to changes in the schedule.

We can however introduce a more formal definition of schedule-robustness: we say that a CL algorithm is $\epsilon$**-schedule-robust** (where $\epsilon > 0$ is a given constant), if

$$\left| \mathcal{L}(S_1(D), f_0, \text{Alg}) - \mathcal{L}(S_2(D), f_0, \text{Alg}) \right| \leq \epsilon, \tag{20}$$

holds for any dataset $D$ and any two schedules $S_1$ and $S_2$ over it. This definition can be weakened to the form

$$\mathbb{E}_{D \sim \mathcal{D}} \, \mathbb{E}_{S_1, S_2 \sim \mathcal{S}} \left| \mathcal{L}(S_1(D), f_0, \text{Alg}) - \mathcal{L}(S_2(D), f_0, \text{Alg}) \right| \leq \epsilon, \tag{21}$$

namely, considering schedule-robustness in expectation with respect to a family of datasets $D$ sampled from a dataset-generating distribution $\mathcal{D}$, and of sequences $S_1, S_2$ sampled (independently) from a sequence-generating distribution $\mathcal{S}$. The latter would capture settings where not all datasets and sequences are possible, but we have prior-knowledge of what kind of sequences and data the CL algorithm might encounter.

While the two above are more rigorous definitions of schedule-robustness, in the present work, we preferred focusing on the more intuitive perspective on the problem, loosely captured by the approximation in (6) with the symbol $\approx$. This is because precisely quantifying the the $\epsilon$ error incurred by SCROLL would require substantial theoretical analysis of the effects of experience replay (a problem akin to algorithmic stability, see e.g. Bousquet & Elisseeff (2002)), which is outside the scope of this work and would not add to our overall take-home-message. We care to point out, however, that our experiments in Sec. 4 show that SCROLL is -schedule-robust with a small $\epsilon$, according to the above definition (at least on $k$-split schedules and Gaussian schedules). This is particularly evident from Fig. 2. Our method is also provably invariant to the set of schedules when all samples from one or more classes is observed together (see discussion in Sec. 3.3). SCROLL's performance does not change as we change the schedule, in contrast to previous methods.

