# OpenReview forum: "Schedule-Robust Online Continual Learning"
_ICLR.cc/2023/Conference — Submitted to ICLR 2023_

### Official Review · Reviewer_wm7u · 2022-10-24

**Confidence:** 3
**Correctness:** 2
**Technical Novelty And Significance:** 2
**Empirical Novelty And Significance:** 2
**Recommendation:** 3

**Clarity, Quality, Novelty And Reproducibility:**

The paper is clear and I did not find technical errors or other details that might be a problem for comprehension or reproducibility.

**Strength And Weaknesses:**

**Strengths:** The problem considered is relevant for the community, as learning from streaming data might be affected by the permutations of the data, that in the paper are denoted as schedules. This problem is well introduced, and the proposed predictor based on combination is meaninful. I particulary liked the idea of the feature extractor that is invariant to permutations for later training the classifier on a different way. Even simpler, the results were understandable and clear to me, with several related baselines that produce a broad picture of the performance.

**Weaknesses:** From my perspective, there are two main points of weakness that make the paper borderline to me. The first one is around the presentation of the schedule and the treatment of such concepts. In the end, when having some i.i.d data, there are already significant contributions and studies of exchangeability and permutation invariant models. For instance, in Bayesian inference, the log-marginal likelihood is invariant to the permutations of the data, and there are multiple ways of building this loss as a combination of log-predictive conditionals where the order and size of batches might change. In the same direction, there are multiple statistical operators that already consider this problem and is somehow a well-studied problem (independent of the continual learning scenario). Therefore, I see important connections missing in section 3.1 when making references to schedules and permutation invariant feature extractors without any reference to well-known concepts in statistics. For example, Eq. (7) seems just a correction of the expectation equation in Eq. (6).

The second point is around the novelty. Even if the schedule robust predictor with the new feature extractor would be well connected to statistics and the mentioned perspectives, the use of Experience replay for the classifier does not seem particularly new w.r.t. (Chaudry 2019). In that sense, I would like to know extra details on which are the exact differences between SCROLL and the previous methods based on Experience replay. Otherwise, the technical contribution is perhaps a bit reduced in my opinion.

**Summary Of The Paper:**

The paper proposes a new method for continual learning from non-stationary data under changing schedules (presentation/observation order of data over time). The general idea is to make the continual learning algorithm robust to different changes of schedules, that is, under exchangeability. For that purpose, the method includes a predictor based on composition. While the first one (or feature extractor) is invariant to permutations (schedules) and trained in an online manner, the second uses Experience Replay (Chaudry 2019).

**Summary Of The Review:**

An interesting paper with two important flaws that make it somehow weak for acceptance.

---

> ### Author Response · Authors · 2022-11-08
> **Response to Reviewer wm7u**
>
> We thank the reviewer for the comments. We address the main questions below.
>
> **Missing Related Works**
>
> Firstly, we kindly ask the reviewer if they could provide key references on permutation-invariant models, such that we can include them in our discussion.
>
> We note however, that the relation with permutation-invariant models **does not undermine the main contribution of our work**, which is to introduce and study schedule-robustness for CL. In particular, other permutation-invariant models, if suitable, could be used as drop-in replacement for NCC or Ridge in SCROLL, since the specific choice of schedule-invariant classifier does not affect our overall proposed strategy.
>
> **Clarification on continual learning scenario**
>
> From the review, it appears that the permutation-invariant models that the reviewer has in mind are designed for “i.i.d data”. However, the key challenge of continual learning is that the data distribution is non-i.i.d. (and possibly adversarial). Without references, it is unclear whether these models would still be permutation-invariant in continual learning settings.
>
> **Clarification on our method**
>
> We have some difficulty understanding the reviewer’s summary of our method. We will try to further clarify our method below.
>
> 1. Our model is a standard ResNet. It is only conceptually split into a classifier and feature extractor for theoretical analysis, and to specify which part of the model is being updated.
> 2. While the feature extractor is (temporarily) fixed and thus technically schedule-invariant, our goal in Sec 3.1 is to show that the classifier (NCC or Ridge regression) is schedule-invariant.
> 3. Experience replay is not a standalone method in our work. Experience replay is used to adapt the entire model (including classifier and feature extractor) $f_T$ learned from Sec 3.1 to $f^*$.
>
> **Novelty regarding Experience Replay**
>
> Compared to the previous methods, our experience replay strategy differs by:
>
> 1. It is performed **after** online learning of current data, instead of learned **along with** current data (see Eq. 3 and its discussion).
> 2. It uses exemplar selection instead of random sampling. We also contributed an explanation to why the former should be favored (see Appendix B.3 and B.4).
> 3. during exemplar selection, we uses a fixed feature extractor, while existing methods all use one continuously updated by the current data.
>
> We note that experience replay is the umbrella term for many individual methods [1, 2]. Small design decisions, as discussed above, matter for empirical performance. This is clearly shown in our experiments, where all methods use some form of experience replay but with vastly different performance. SCROLL outperforms existing methods by a large margin.
>
> **Novelty and significance**: For a full discussion on novelty, contribution and significance, please see [our general comment](https://openreview.net/forum?id=bkxynaG3Vm7&noteId=bqUV0yp_DA).
>
> [1] Rebuffi et al., “icarl: Incremental classifier and representation learning”. CVPR, 2017
>
> [2] Buzzega et al., “Dark experience for general continual learning: a strong, simple baseline”, NeurIPS 2020

---

### Official Review · Reviewer_5vq1 · 2022-10-24

**Confidence:** 3
**Correctness:** 3
**Technical Novelty And Significance:** 2
**Empirical Novelty And Significance:** 2
**Recommendation:** 5

**Clarity, Quality, Novelty And Reproducibility:**

Overall the paper was easy to read. Regarding quality, in my opinion schedule-robustness is not formally well-defined (see comment above)  and should become more rigorous mathematically. With respect to novelty I have the impression that the current paper is a small incremental contribution on top of recognising that some pre-existing approaches satisfy some desired property. Finally, I don't see any reason why the empirical results should not be reproducible.

**Details Of Ethics Concerns:**

No ethical concerns.

**Strength And Weaknesses:**

The contributions of the paper are threefold:

1. The notion of schedule-robustness is introduced (see Equation 5). Although I do intuitively understand the definition some things are unclear. First, what is the exact meaning of the $\approx$-sign? Second, should Equation (5) hold on every dataset D, or only on some datasets?

2. It is observed that some specific pre-existing classifiers are already schedule-robust. This gives a specific algorithm called SCROLL

3. Scroll is evaluated experimentally. Not surprisingly, SCROLL outperforms other approaches when they are applied to different schedules than "expected". However it also outperforms them on the schedules for which they were designed, which I don't fully understand. Do you have some intuition on why that is?

**Summary Of The Paper:**

The paper concerns itself with continual learning and observes that current methods are making assumptions on the schedule that can be unrealistic. In order to ameliorate this, the notion of \emph{schedule-robustness} is introduced. Intuitively schedule-robustness for CL means that the performance is "independent" of the schedule. Empirical evaluations are also provided.

**Summary Of The Review:**

Given the comments above, I believe that the paper needs improvements in mathematical rigorousness, and some discussion on why SCROLL outperforms other algorithms even on the schedules that they are designed for, as (at least for me) this does not check out and I am looking forward for the author's response in the rebuttal.

---

> ### Author Response · Authors · 2022-11-08
> **Response to Reviewer 5vq1**
>
> We thank the reviewer for the comments. We address the main questions below.
>
> **Definition of schedule-robustness.**
>
> We first clarify the definition of schedule-robustness. Ideally, we wish to achieve schedule-invariance, where $\approx$ in Eq. 5 can be replaced by the $=$ operator. Schedule-invariance means that the model is truly “independent” of the schedule, achieving identical performance under all schedules and data. As an example, ridge regression (Sec. 3.1) is schedule-invariant. The proposed model adaptation (Sec 3.3) is schedule-invariant when all the samples of one or more classes are presented at once. Eq. 5 is meant to hold for **all** datasets.
>
> In practice, however, some randomness may occur during learning (e.g., replay samples selection when data comes in very small batches). Due to these factors, we may achieve similar but not identical performance under different schedules. In this sense, schedule-robustness aims to weaken the invariance constraint by allowing Eq. 5 to be approximate. Formally, one could introduce a notion of $\epsilon$-schedule-robustness, requiring the performance of the same model trained on any two different schedules to be different only up to some $\epsilon>0$ error. This can be written as $|\mathcal{L}(S_1(D), f_0, Alg) - \mathcal{L}(S_2(D), f_0, Alg)| \leq \epsilon$ holding for any dataset $D$ any two schedules $S_1, S_2$ (or, from a meta-learning perspective, in expectation with respect to the family of datasets and schedules that we could face in practice).
>
> While this is a more rigorous definition of schedule-robustness, in the present work, we preferred focusing on a more intuitive perspective on the problem, loosely capturing the approximation in Eq. 5 with the symbol $\approx$. This is because precisely quantifying the $\epsilon$ error for SCROLL would require substantial theoretical analysis of the effects of experience replay (a problem akin to algorithmic stability [1]), which would be outside the scope of this work and would not add to our overall take-home-message. We care to point out, however, that our experiments show that SCROLL is $\epsilon$-schedule-robust with a small $\epsilon$, according to the above definition (tested on k-split schedules and guassian schedules). Our method is also provably invariant to the set of schedules when all samples from one or more classes is observed together. SCROLL’s performance does not change as we change the schedule, in contrast to previous methods.
>
> We thank the reviewer for the question. We will add a discussion to better convey what we intend by schedule robustness, providing an intuition of what would be needed to derive a more formal definition, as noted above.
>
> **Outperforming other methods on standard schedules**
>
> For all schedules (including standard ones)**,** SCROLL outperforms baseline methods because:
>
> 1. SCROLL is better at mitigating forgetting. As discussed in Sec 3.1, achieving schedule-robustness via NCC or Ridge regression also mitigates forgetting. This is one of the main factors in degrading the performance of CL algorithms.
> 2. Baseline methods were not designed to fully exploit pre-training [2]. When combined with forgetting, this implies that the benefits of pre-training fade quickly. In contrast, SCROLL builds on top of the meta-learning interpretation to fully exploit pre-training.
>
> To further validate our results, we care to point out that, for existing methods:
>
> 1. We performed extensive hyper-parameter tunings to ensure fair comparison (see Appendix D).
> 2. Performance results reported for them are **higher** than previously reported due to the benefits of pre-training.
>
> **Novelty and contribution**
>
> We note that our method is more than identifying “some specific pre-existing classifiers that are already schedule-robust”. In particular, we incorporated experience replay in our method while maintaining schedule-robustness (Sec 3.3). Our key technical contribution is the entire CL pipeline, which we also show both theoretically (for a class of schedules) and empirically to be schedule-robust.
>
> **Empirical Significance**
>
> We are also surprised by the *“marginal empirical novelty and significance”* score attributed to our empirical significance.
> Our experiments systematically demonstrated 1) that sensitivity to schedule is a significant issue in previous CL algorithms and 2) that SCROLL outperforms the previous methods by a large margin (e.g., 20% in many settings).
>
> For a full discussion on our contribution and novelty, please see [our general comment](https://openreview.net/forum?id=bkxynaG3Vm7&noteId=bqUV0yp_DA).
>
> [1] O. Bousquet and A. Elisseeff. Stability and generalization. J. Mach. Learn. Res., 2:499–526, 2002.
>
> [2] Sanket Vaibhav Mehta et al., “An empirical investigation of the role of pre-training in lifelong learning”. arXiv preprint, 2021.

---

> > ### Comment · Reviewer_5vq1 · 2022-11-25
> > **Thanks**
> >
> > Thanks for the extensive response.

---

> > > ### Author Response · Authors · 2022-11-25
> > > **Reply to Reviewer 5vq1**
> > >
> > > We kindly thank the reviewer for acknowledging our response.
> > >
> > > Could you please clarify whether we managed to address your concerns?
> > >
> > > If not, could you please elaborate on any outstanding issues? We are happy to further clarify.

---

> > > > ### Comment · Reviewer_5vq1 · 2022-11-25
> > > > **-**
> > > >
> > > > I still am of the opinion that the rigorous definition is important to include in the paper rather than the vague one. As I mentioned already in my review, some parts in the paper were unclear because of it so I disagree that this is a more intuitive perspective.
> > > >
> > > > But I understand your point and in the end this is perhaps a matter of taste. I don’t have any additional comments/outstanding issues to add.

---

> > > > > ### Author Response · Authors · 2022-11-25
> > > > > **Reply to Reviewer 5vq1**
> > > > >
> > > > > #
> > > > >
> > > > > We thank the reviewer for the additional feedback.
> > > > >
> > > > > We agree that a formal definition is important. This is why, in the revised version (16 Nov)**, we discussed the distinction between schedule-invariance and -robustness (see Sec. 2.2 in blue) and provided a formal definition of $\epsilon$-schedule-robustness in Appendix E.**
> > > > >
> > > > > The reviewer’s concern has a straightforward solution: we will move this formal definition to the main text in the final version of the paper.
> > > > >
> > > > > We stress that this new definition doesn't change the take-home message of our paper: our analysis shows that SCROLL is schedule-invariant for a family of schedules and, empirically, schedule-robust for general schedules.
> > > > >
> > > > > Since the reviewer stated *"[...] don’t have any additional comments/outstanding issues to add"*, we believe that we have sufficiently addressed the other concerns, including one regarding contribution and significance.
> > > > >
> > > > > - Would the reviewer consider re-evaluating the score on empirical contribution and significance, given our previous response?
> > > > > - Further, if we have addressed all the concerns, could the reviewer consider an overall re-evaluation of our work?

---

### Official Review · Reviewer_dgx2 · 2022-10-25

**Confidence:** 4
**Correctness:** 3
**Technical Novelty And Significance:** 2
**Empirical Novelty And Significance:** 3
**Recommendation:** 5

**Clarity, Quality, Novelty And Reproducibility:**

Clarity: They could make a fair amount of the paper a lot more concise and to the point. It would be really helpful to clearly point out what is their contribution and what isn’t (ex. reveal beforehand things like the Buffering strategy (3.3) are based on someone else’s work, it just felt like that fact was revealed a bit too late; there are a lot of details in section 3 which could be moved to the sections of the related work.). If they talk about their work more directly from the perspective of making the memory buffer robust to schedules, it might make things more intuitive to understand (there are a lot of other ways one could make CL robust to schedule, and from the paper’s headline I was hoping to see something a lot different).

Quality/Novelty: Their approach seems very procedural. Not much theoretical technical novelty. Otherwise, a good amount of experiments and comparisons against other approaches.

Reproducibility: Seems fairly easy to reproduce. However, we don't have the exact details of the splits, so getting the same exact numbers at the output won’t be possible. No code or data was provided.


**Strength And Weaknesses:**

Strengths:
- Thorough and clearly explained experiments where they improve upon the SOTA.
- Marginally novel technical contribution.
- Increasing the memory buffer size increases the curriculum learning accuracy. However, Residual Adapters helped get better performance at lower buffer sizes than the full model training. And the ridge regression initialization did the best job. Another way to think about this work is to aim toward making the memory buffer robust to the schedule.

Weaknesses:
- Only used on CIFAR10,100 with the help of resnet 18. It would help to see the results on other datasets and networks as well. We need to make sure that the results are data/model agnostic.
- The transition from F_0 -> F_t -> F_T-> f* is very confusing.
- They need to do a better job at stating what is being trained and what isn’t and what losses are being used at each stage. Maybe having a better diagram or a few more diagrams or just a table explaining the different stages and loss combinations of training will help.
- It wasn’t very clear if the memory buffer is big, does the whole ψ get trained or not??
- Additionally, they need more experiments like Fig.2 (The paper is about preventing catastrophic forgetting, and I only see one experiment where they are analyzing the model’s performance across time).
- It also isn’t clear whether Fig2. is using NCC or Ridge regression.
- They need to clearly state as bullet points or a list all the factors which are contributing towards making their method schedule robust. It felt like these factors were all over the place.

**Summary Of The Paper:**

This work builds on the prior work of curriculum learning to make the curriculum learning process neutral to the training schedule. They first form initial pre-representation-training of the network with the help of NCC/ridge regression. The network is divided into 2 components: ϕ (classifier) and ψ (backbone). The ψ from the pre-training is used for the further training of the network (ϕ is discarded). ψ stays fixed after this (This combined with NCC or the ridge regression are the components that help make the entire training process neutral to the training schedule).

This is followed by the part where they perform online curriculum learning. They employ memory buffers and episode replay techniques for this purpose. For a given input batch from a data stream, they update the memory buffer and the NCC/Ridge-regression clusters accordingly. Here they also update the ϕ/classifier parameters. Additionally, if the memory buffer is small, they also have some residual connections from ψ to ϕ which also get updated.

**Summary Of The Review:**

A good attempt at solving a not-so-well-studied problem of curriculum learning. As stated before, marginal technical novelty. I personally couldn’t find a comparable schedule-invariant curriculum learning approach. However, they could have done a better job at making the paper’s theory simpler, easier, and most importantly, intuitive to read. Their work improves over SOTA or was very comparable and delivered what they promised. Therefore, at the moment I am leaning toward a marginal rejection.

---

> ### Author Response · Authors · 2022-11-08
> **Response to Reviewer dgx2**
>
>
> We thank the reviewer for their comments. We address a few misunderstandings that emerged from the review and clarify our work below.
>
> **Curriculum vs Continual Learning.**  We would like to clarify that our work focuses on continual learning rather than curriculum learning, as noted by the reviewer. While the notion of schedule may resemble that of a curriculum, a key substantial difference is that schedules cannot be controlled/chosen in continual learning and might even be adversarial.
>
> **Contribution.** Our work is not (only) about *“making the memory buffer robust to the schedule”* but rather about making the entire CL pipeline schedule-robust. To achieve schedule-robustness, our method relies on a) schedule-invariance from online learning (NCC or Ridge), b) performing model adaptation only after online learning, and c) making the memory buffer robust to schedules. Please see Sec 3.1 and 3.3 for a detailed discussion.
>
> Therefore, we respectfully disagree with the reviewer’s opinions *“a lot of details in section 3 which could be moved to the sections of the related work”* and *“Not much theoretical technical novelty”* since Section 3 contains **important contributions related to the novel concept of schedule-robustness.** In particular, Sec 3.1 shows how ridge regression and NCC are schedule-robust, which is necessary to show that our entire algorithm is schedule-robust in Sec 3.3. Finally, Sec 3.2 adds to our theoretical contribution, connecting CL to standard meta-learning via schedule-robustness and in doing so, justifying the use of pre-training in CL.
>
> Please see [our general comment](https://openreview.net/forum?id=bkxynaG3Vm7&noteId=bqUV0yp_DA) for a full discussion on novelty, significance, and contribution.
>
> **Distinction between models and losses.**  Following the reviewer’s suggestion, we will revise our discussion to describe more clearly the role of the components $f_0$, $f_t$, $f_T$, and $f^*$:
>
> - $f_0$ denotes the pre-trained network before learning from the sequence. Both $\phi$ and $\psi$ are trained. Training is performed by minimizing the cross-entropy loss.
> - $f_t$ is the online model obtained after learning from batch $B_t$, starting from $f_0$. Here $\psi$ is kept fixed while $\phi_t$ is updated either via NCC or Ridge regression.
> - $f_T$ is the last online model obtained after observing and learning from the full sequence (**before** model adaptation). Again learned via NCC or Ridge regression.
> - $f^*$ is the final model used for testing. Here both $\phi$ and $\psi$ are trained, adapting $f_T = \phi_T \circ \psi$ to the replay data. This is done by minimizing the cross-entropy loss.
>
> We thank the reviewer for the comment. We will improve the discussion about the above flow in Sec 3.4, Alg. 1 and Figure 1. We emphasize that model adaptation happens **after** online learning, not *“during online curriculum learning”* as summarized by the reviewer. Model adaptation is performed as standard supervised learning, with multiple epochs over the replay data.
>
> **Buffers size and training $\psi$.** Model adaptation updates the feature extractor $\phi$ via either residual adapters or full model training. When the replay buffer size is large, $\phi$ gets updated (see Sec 3.4 full-model tuning and Fig 1, where green-shaded components are updated).
>
> **About Figure 2.** We clarify that the figure does ****not**** show forgetting over time. Rather, it plots the **final** model performance of different methods under different schedules. This is intended to show how robust each method is to changing schedules. Fig 2 uses Ridge regression. To report on SCROLL’s behavior wrt forgetting, we will add a new section and figure in the appendix.
>
> **Additional Datasets and architectures.**  Sec 4.2 describes a miniImageNet experiment using a ResNet-12 architecture. In this case, the pre-training only uses a much smaller dataset (38400 images from the 64 classes of the meta-train split). As it can be appreciated by the successful results in Table 2, our proposed strategy is also applicable to more general settings and datasets. More details about the miniImageNet experiment can also be found in appendix D.2.

---

> > ### Comment · Reviewer_dgx2 · 2022-11-29
> > **Ackknowledgement**
> >
> > I have read and appreciate all the work done by the Authors. A lot of clarifications have been provided, and that is useful. The paper contains a contribution, although this does not lead to a change in the initial rating.

---

> > > ### Author Response · Authors · 2022-11-29
> > > **Seeking Clarification**
> > >
> > > We thank the reviewer for acknowledging our response.
> > >
> > > We feel that we did address your concerns. So could we seek some clarification on what outstanding concerns are for the current score? We are happy to further discuss and understand areas for improvements.

---

### Official Review · Reviewer_kayx · 2022-10-27

**Confidence:** 4
**Correctness:** 3
**Technical Novelty And Significance:** 2
**Empirical Novelty And Significance:** 2
**Recommendation:** 5

**Clarity, Quality, Novelty And Reproducibility:**

The presentation of the paper is relatively clear. I am not still convinced of its novelty, however. Also, I think there might be some implementation details missing from the text.

**Strength And Weaknesses:**

STRENGTHS
- The paper raises the issue of schedule robustness, which is very interesting in my opinion.


WEAKNESSES
- I think the contribution of the paper is very limited. In my opinion, the most difficult aspect of continual learning is representation learning, but the proposed approach assummes a pre-trained feature extractor and only deals with the schedule robustness of the final layer of the model. Moreover, training a classifier online while keeping a pre-trained feature extractor frozen has already been explored in past work [1].
- The use of residual adapters does not really fit into the paper since they do not contribute to schedule robustness. Moreover, we can see from the results in Table 4 that they are only beneficial when the memory size is very very small. Also, it is not possible to say whether they would work well for other kinds of architectures, or even other kinds of convolutional architectures (only ResNet variants are used in the paper).
- Regarding the results in Figure 2, I think that the main factor that allows scroll to outperform the other methods is that the feature extractor is kept frozen.
- A minor comment: there are some formatting issues with the pdf (for example, see top of page 9, Table 3 appearing after Table 4, etc.).


QUESTIONS:
- It is unclear to me how a continual learning setting can be both task-free and class-incremental at the same time. Could you explain further?
- For the results in Tables 1 and 2, do the data arrive in small mini-batches online, or does the model have access to all the data from the present task at the same time?
- I quickly skimmed over the proof that ridge regression is schedule robust and it seems to me that the proof assumes that all data are available at the same time. This assumption does not hold in online continual learning though. Could you please explain further? Could you also give an intuitive explanation of how ridge regression can be schedule robust?
- Why do you think NCC performs worse than ridge regression?
- Is SCROLL applicable to streams with tasks that are not class-disjoint?


REFERENCES
[1] Hayes, T. L., & Kanan, C. (2020). Lifelong machine learning with deep streaming linear discriminant analysis. In Proceedings of the IEEE/CVF conference on computer vision and pattern recognition workshops (pp. 220-221).

**Summary Of The Paper:**

This work raises the issue of schedule robustness in continual learning, and proposes an approach that is schedule-robust by construction. The proposed approach assumes access to a pre-trained feature extractor, then trains a one-layer classifier online, and adapts the feature extractor after the end of the stream using only memory data.

**Summary Of The Review:**

Despite the fact that the paper raises an important issue in continual learning (i.e., schedule robustness), I am not convinced that its contribution is significant enough for a top conference like ICLR.

---

> ### Author Response · Authors · 2022-11-08
> **Response to Reviewer kayx 1/2**
>
> We thank the reviewer for their comments. We address the main points raised by the reviewer below. \
> Please also see [our general comment](https://openreview.net/forum?id=bkxynaG3Vm7&noteId=bqUV0yp_DA) for a full discussion on novelty, significance, and contribution.
>
> **Scroll does not perform representation learning?** It does. We would like to clarify SCROLL *performs representation learning* by updating the feature extractor during model adaptation (Sec. 3.3). Moreover, the online classifier serves as initialization and is jointly adapted with the feature extractor. According to our empirical results, such model adaptation is vital for the final performance (see $f_T$ vs $f^*$ in Table 3).
>
> **Is SCROLL schedule-robust in the final layer **only**?** No, the schedule-robustness property applies to the entire model. **First** we learn a schedule-invariant classifier (Sec 3.1), **then** we learn a schedule-robust representation and classifier using replay data only (Sec 3.3).
>
> **Connection with (Hayes & Kanan 2020).** We thank the reviewer for the new reference (Hayes & Kanan, 2020). We will include it in our discussion. However, we highlight two technical differences in our method:
>
> 1. Our linear classifier (NCC/Ridge) is strictly schedule-invariant, while the method proposed by Hayes & Kanan, 2020 is *“sensitive to class ordering”* (see the last paragraph of Sec 3 in Hayes & Kanan, 2020).
> 2. Our method incorporates experience replay, which is key to adapt the feature representation. This was identified as a challenge and future work in Hayes & Kanan, 2020.
>
> **Relevance of Residual Adapters.** To ensure schedule-robustness, we rely on model adaptation using ****only**** replay data. Since the total amount of replay data can be limited in many applications, residual adapters are an effective solution to prevent overfitting (in contrast to fine-tuning) while still improving upon f_T (which would otherwise keep the representation fixed and only learn the last layer). In particular, we see that full model training with small buffer data performs worse than simply learning the linear layer (see Table 4).
>
> **Residual Adapters for other Architectures.** The adoption of residual adapters is motivated by the empirical observation that partial model updates perform better when training data is limited. This idea has been extensively studied and shows good performance for state-of-the-art architectures, including ResNet [1] and transformers [2, 5]. Extending our method to other architectures is an interesting direction that we discussed as future work, but beyond the current scope.
>
> **Why does SCROLL perform well?** Given the discussion above, we respectfully disagree with the reviewer’s opinion that *“[...] the main factor that allows SCROLL to outperform the other methods is that the feature extractor is kept frozen”*. In particular, we emphasize that:
>
> 1. Our feature extractor is actually updated (not kept frozen). Empirically, this is important (see Table 4).
> 2. Counterexample to the reviewer’s statement: on CIFAR-10, GDumb with a 2000-buffer achieves 87.3% (Table 1), while SCROLL with fixed-representation (i.e. f_T) only obtains 81.4% (Table 3). This shows that fixed-representation is not sufficient. We will add the fixed-representation variant to Table 1 to stress this point.
> 3. The frozen feature extractor by itself is not a solution for continual learning. The choice of the linear classifier is also important for mitigating forgetting and achieving good performance. We intentionally chose NCC and ridge regression for their schedule-invariance, rather than logistic regression which is the standard formulation for classification problems.

---

> > ### Author Response · Authors · 2022-11-08
> > **Response to Reviewer kayx 2/2**
> >
> >
> > **Detailed Questions**
> >
> > We address here the detailed questions the reviewer
> >
> > 1. **Task-free and Class Incremental settings.** We refer to [4] (see also [3]) for a thorough discussion on the distinction between task-free, task-based, and class incremental settings. In short, class-incremental describes a setting where a single, unified classifier layer is learned for all classes (a.k.a. “single-head”). This is in contrast with task-based settings where each task could require learning a separate classifier layer focused only on the classes within each task.
> > In task-free settings, data may come in small batches with no clear task distinction. Class-incremental is, therefore a special case of this setting where a unified classifier layer is learned for all data batches.
> >
> > 1. **Data schedules in the experiments.** We allow baseline methods to follow their intended setups. Offline methods access a large batch containing all the samples from one or more classes, while online methods (including ours) access mini-batches one at a time. We also stress that **our method can handle both settings.**
> > 2. **Details on our Proof.** In our proof, we showed that the closed-form solution for ridge regression consists of two terms that can be expressed as summations. Importantly, **the summations can be computed online** (Eq. 9) and is equivalent to the batch solution in the appendix. In addition, the summation is invariant to the data order, which achieves schedule invariance.
> > 3. **NCC vs Ridge.** In contrast to NCC, Ridge regression keeps track of both the first and second moments of the data: NCC learns only the class prototypes (their mean/first moment). In contrast, Ridge regression also learns a change in metric based on the inverse covariance matrix (encoding the second moment of the data) of the input features (contrast Equations 7 and 9). We may loosely interpret this transformation as suppressing high-variance features and amplifying stable features for all classes, improving the robustness of the learned model.
> > 4. **Non class-disjoint tasks.**  SCROLL is applicable to tasks with non-disjoint classes. Moreover, SCROLL embodies our goal of achieving schedule-robustness. In particular, it does not make *any* assumptions about the way data is presented. Each batch can, for example: 1) contain data from any number of classes, 2) contain a different distribution of class samples, 3) contain new data from a class observed many batches back in the past, mixed with data from a new class, etc. For instance, see also our experiment with a gaussian schedule in appendix B.2.
> >
> > [1] Perez, Ethan, et al. "Film: Visual reasoning with a general conditioning layer." AAAI 2018.
> >
> > [2] M. Sandler, A. Zhmoginov, M. Vladymyrov and A. Jackson, "Fine-tuning Image Transformers using Learnable Memory," *CVPR*, 2022
> >
> > [3] Rahaf Aljundi et al. “Task-free continual learning”, CVPR 2019
> >
> > [4] Zheda Mai et al., “Online continual learning in image classification: An empirical survey.” Neurocomputing, 2021
> >
> > [5] Neil Houlsby et al., “Parameter-Efficient Transfer Learning for NLP”, ICML 2019

---

> > > ### Comment · Reviewer_kayx · 2022-12-13
> > > **Reply to Authors**
> > >
> > > Dear authors,
> > >
> > > I would like to (very belatedly) thank you for your reply. I also want to sum up what I think are the main weaknesses of this paper.
> > > - This paper assumes access to a good pre-trained feature extractor, and only updates it via adapters after all data have been observed. As I wrote in my review, I consider continual representation learning to be the most important issue in continual learning. Moreover, I think that, in practice, we would like the model to be able to perform inference during the continual learning process (instead of only after all data have been seen). Therefore, since SCROLL only updates the feature extractor after all data have been seen, it would not be a valid solution if a pre-trained feature extractor was not available and we had to start with a randomly initialized feature extractor.
> > > - There is some overlap with previous work. The approach proposed by Hayes & Kanan, 2020, is only sensitive to the class oredering when the covariance matrix is not fixed. In the case where the covariance matrix is fixed to the identity matrix, the approach becomes insensitive to class-ordering, hence somewhat similar to what you have proposed.
> > > - The use of adapters is irrelevant to the main theme of the paper. In particular, while the use of adapters does improve the final accuracy of the model, it does not make it less sensitive to the data schedule. I am sure you would agree that SCROLL is equally insensitive to the data schedule regardless of whether adapters are used or not.
> > >
> > > In short, the only contribution the paper makes is the introduction of the schedule-robustness problem. (I am not sure whether your proof about ridge regression is novel or not.) I think the paper would be significanlty strengthened if you were to find a way to train the feature extractor online in a schedule-robust way.

---

> > > > ### Author Response · Authors · 2022-12-13
> > > > **Further clarifications**
> > > >
> > > > We thank the reviewer for acknowledging our response. We wish to make several further clarifications.
> > > >
> > > > **Continual Representation Learning.** We would like to clarify that actually SCROLL is capable of continual representation learning (i.e., updating the feature extractor) during the continual learning process. We discussed this in both the original (Sec 3.4 intermediate predictors) and revised version (Appendix B.5). Please see also Appendix B.5 for further details and additional experiments on intermediate predictors.
> > > >
> > > > **Relevance of the residual adapter.** As we discussed in our original response, SCROLL maintains schedule-robustness by updating the representation using **only replay data**. With limited replay buffers, residual adapters are necessary to prevent overfitting and improve model performance.
> > > >
> > > > We clarify that while schedule-robustness is a key property for continual learning, model performance is also a key metric. To this end, we have shown our method empirically outperforming various baselines by a large margin while maintaining schedule-robustness.
> > > >
> > > > **Comparison to Hayes & Kanan, 2020.** Our method is schedule-robust rather than just insensitive to class-ordering. This is a significant difference. Additionally, “setting” the covariance matrix to be identity yields an estimator more similar to our NCA formulation than ridge regression. We have shown that NCA performs significantly sub-par in contrast to ridge-regression (for which using a non-trivial covariance is key). We refer to our ablation experiments.
> > > >
> > > > We would like to highlight, again, that while schedule-robustness is a key concept we introduce in this work, another key contribution of our work is to show SCROLL’s superior empirical performance.
> > > >
> > > > **Access to pre-trained representation.** We highlight several reasons for using the pre-trained representation:
> > > >
> > > > 1. For most real applications, we argue that it would be **unrealistic to \*not\* use a pre-trained representation** (e.g. deploying randomly initialized models to directly train from scratch on customer’s data in computer vision). Even more so, since off-the-shelf pre-trained models are readily available for the vast majority of application domains (e.g., NLP), adapting them for specific tasks/problems is becoming increasingly evident as a very effective strategy.
> > > > 2. Our method updates the pre-trained representation from the streaming data and demonstrates improved performance with the updated representation.
> > > > 3. Our comparison is fair. All baseline methods are given the same pre-trained representation.
> > > > 4. Pre-trained representation helps improve all tested methods, including the baselines. We also presented a meta-learning perspective to justify pre-training (Sec. 3.1 and 3.2), another contribution of our work. Given these observations, we discussed further investigating pre-training for continual learning in Sec 5.
> > > >
> > > > **Contributions** We emphasized schedule-robustness in our work since it is a key and novel concept.
> > > >
> > > > However, as discussed in our previous response and above, our contributions are much more than that: our method is capable of continual representation update **and outperforms various baselines by a large margin**. We also presented principled justification for adopting pre-training for continual learning.

---

### Author Response · Authors · 2022-11-08
**To All reviewers: on novelty, contributions and significance of our work**

We thank the reviewers for their feedback and valuable time. Please see both our detailed answers to reviewers’ specific questions/comments and our main discussion on novelty below.

We are confident that our clarification will provide a fresh perspective on both our technical and conceptual contributions and will serve as a springboard for a meaningful discussion on the re-evaluation of our work.\
Should the reviewers have any questions, we will be happy to engage in the discussion.

&nbsp;
&nbsp;
&nbsp;

### Clarification on novelty, contributions, and significance

The reviewers found our work lacking in technical novelty. However, we emphasize that our work primarily differentiates itself by **conceptual novelty**, substantiated by both theoretical and empirical contributions, which we will also highlight below.

**Conceptual novelty.**

A key contribution of this work is to introduce and study the notion of “schedule-robustness”.
We note that all reviewers found this to be interesting and novel.

In this sense, we highlight that *our contribution is conceptual rather than technical*, since achieving schedule-robustness is a general goal. Concretely, we showed both theoretically and empirically that practical and sound strategies exist to achieve schedule-robustness. Our strategies are modular and flexible, thus opening up new directions for future CL research (as we comment in Sec. 5).

The conceptual novelty of our strategy is nicely captured by reviewer **dgx2**’s comment, *“I personally couldn’t find a comparable schedule-invariant curriculum learning approach.”*

**Technical contribution.**

While our resulting algorithm leverages well-established components (NCC, Ridge, experience replay), the main technical contribution is the resulting CL pipeline to achieve schedule-robustness. The pipeline organizes CL into two independent stages: 1) online classifier learning followed by 2) model adaptation with replay data only. To our knowledge, this differs from all existing approaches.

**Theoretical contribution.**

We wish to highlight a contribution that some reviewers might not have fully appreciated, namely that our strategy is principally motivated by theoretical analysis:

1. We have justified the use of pre-training in CL from a meta-learning perspective. This allows us to better understand the connection between CL and meta-learning (e.g., identifying that ProtoNet is naturally a schedule-robust continual learner). We also justified why multi-class classification is a principled way for performing pre-training (see Section 3.2)
2. We showed that our method is “schedule-invariant” - a stronger property than schedule-robustness - when one or more classes are presented at once.

**Empirical Significance.**

We are honestly surprised by the score *“marginal empirical novelty and significance”* that most reviewers attributed to our work. On the empirical front, our contributions are significant:

1. We systematically demonstrated that many existing methods have drastically degraded performance when schedule changes. Even subtle changes (e.g., using different splits) can cause such drops. For instance, for a sequence of classes A, B, C, D, E, F, presenting them as (A, B), (C, D), (E, F) versus (A, B, C) and (D, E, F) often leads to very different performance, *even when the order of the classes remain the same*. In particular, the performance degradation appears more severe when dividing a large number of classes into fine splits. This is a significant issue for existing CL methods – see Fig 2 for an analysis of the effect of different schedules.

2. In contrast with existing methods, **SCROLL achieves schedule-robustness**. This allows us to outperform existing methods by a large margin. In CIFAR-100, the difference in test performance **is over 20%** (see Fig 2 and Table 5 in the appendix) under different buffer sizes and splits in CIFAR-100. In CIFAR-10 and miniImageNet, we also **outperform baselines by an average of 5 to 10%**. These performance improvements are clearly significant.

**We would kindly ask the reviewers whether they agree with the points above or suggest possible improvements, if not.**

---

### Author Response · Authors · 2022-11-15
**To all Reviewers: uploaded revision changes**

We thank the reviewers again for their comments.

We have updated our manuscript to reflect the reviewers’ suggestions.\
The **updated segments are all colored blue** for easier comparison with the original version.

We also include a summary of changes below:
1. A revised description of our contributions and their significance.
2. Added (Hayes & Kanan 2020) as a reference with discussion.
3. Added results for $f_T$ to Table 1 for easier comparison.
4. Added the experiment for showing the evolution of generalization performance to visualize forgetting in appendix B.5 (see also Figure 5).
5. Revised the definition of schedule-robustness and added a related discussion in appendix E.
6. Layout fix such that now Table 4 comes after Table 3.
7. Specified SCROLL version (Ridge) in figure 2.

**We look forward to any further feedback from the reviewers and are happy to engage in the discussion.**

---

### Decision · Program_Chairs · 2023-01-20

**Decision:**

Reject

**Justification For Why Not Higher Score:**

Please see above.

**Justification For Why Not Lower Score:**

NA

**Metareview: Summary, Strengths And Weaknesses:**

The paper presents a continual learning algorithm that is schedule invariant: the order with which new data is presented has no effect in the trained classifier distribution. The paper introduces this concept, which has not been studied in the past, certainly not in the continuous learning setting.  In short, the scheme uses a pre-trained (over offline data) model as a feature extractor, and trains a shallow classifier on top of it using a sample-data permutation invariant learning method (such as ridge regression). The extractor is then fine-tuned using a rehearsal based/experience replay use of the new data. The proposed method is shown to outperform several competitors in final accuracy, while also exhibiting robustness.

This is an interesting contribution, introduces a new concept, and comes with a rather extensive set of experiments to support the proposed claims. That said, several issues remain post-rebuttal.
1) Continuous learning,  both task free and task based, assumes that classes come batched for a reason: predictive performance is supposed to be good at *intermediate points*, at the conclusion of each task. This is why accuracy is reported at the end of each task, and performance is measured in the end via average accuracy (averaged across tasks). The motivation is that the model should operate "in the field" in between tasks, and adapt whenever new tasks arrive, as needed. Understanding behavior under an arbitrary schedule is interesting, but measuring performance only at the end (at T, as indicated by the present paper) changes the setting-and the bar with which competitors are measured-a bit. It could be argued that what matters is performance at the end (all tasks revealed/testset coming from all new tasks), but then the algorithm is not exactly "continual": it is not clear how the proposed method performs at intermediate points, or equivalently, at the next change in the environment. It is even not clear if it would the proposed algorithm is schedule robust in this case, due to the adaptation of the feature extractor: if we did introduce intermediate accuracy measurement points, would the feature extractor be adapted at these points? Would it be kept fixed throughout?  All of this is a bit swept under the rug by the definition of a schedule, schedule robustness, and the focus on final accuracy. Note that this issue is distinct from whether the ridge regression solution can be obtained incrementally.
2) It is clear that the ridge regression classifier is schedule invariant, but it is not clear that the entire process, including the end-to-end retraining of the feature extractor, is schedule robust: this is not formally proved. Doing so would require using the formal definition given by the authors in the appendix, during the rebuttal-it is good this was introduced. If the main contribution is conceptual (the introduction of robustness), this would need to be argued beyond experimentation. This is also because (a) the technical contribution of using ridge regression etc. is limited (emphasis on the technical here, the reviewers appreciated the conceptual novelty) and (b) it is not clear how such a formal claim would interact with issue 1 (the requirement that the learning is continual). For example, should robustness hold only at T? What does this mean, an how should it even be defined, if accuracy also matters at intermediate points?




**Summary Of Ac-Reviewer Meeting:**

NA